# Towards a generic prototyping approach for therapeutically-relevant peptides and proteins in a cell-free translation system

Yue Wu[1,2,15], Zhenling Cui[3,4,5,6,7,8,9,15], Yen-Hua Huang [1,10], Simon J. de Veer [1,10], Andrey V. Aralov [11], Zhong Guo[3,4,5,6,7,8,9], Shayli V. Moradi[3,4,5,6,7,8,9], Alexandra O. Hinton [1], Jennifer R. Deuis[1], Shaodong Guo [1], Kai-En Chen [1], Brett M. Collins [1], Irina Vetter[1,12], Volker Herzig [1,13,14], Alun Jones[1], Matthew A. Cooper [1], Glenn F. King [1,10], David J. Craik [1,10], Kirill Alexandrov [3,4,5,6,7,8,9✉] & Sergey Mureev [5✉]

Advances in peptide and protein therapeutics increased the need for rapid and cost-effective polypeptide prototyping. While in vitro translation systems are well suited for fast and multiplexed polypeptide prototyping, they suffer from misfolding, aggregation and disulfide-bond scrambling of the translated products. Here we propose that efficient folding of in vitro produced disulfide-rich peptides and proteins can be achieved if performed in an aggregation-free and thermodynamically controlled folding environment. To this end, we modify an *E. coli*-based in vitro translation system to allow co-translational capture of translated products by affinity matrix. This process reduces protein aggregation and enables productive oxidative folding and recycling of misfolded states under thermodynamic control. In this study we show that the developed approach is likely to be generally applicable for prototyping of a wide variety of disulfide-constrained peptides, macrocyclic peptides with non-native bonds and antibody fragments in amounts sufficient for interaction analysis and biological activity assessment.

[1] Institute for Molecular Bioscience, The University of Queensland, Brisbane, QLD 4072, Australia. [2] Cold Spring Harbor Laboratory, Cold Spring Harbor, New York, NY 11724, USA. [3] ARC Centre of Excellence in Synthetic Biology, Sydney, VIC, Australia. [4] Centre for Agriculture and the Bioeconomy, Queensland University of Technology, Brisbane, QLD 4001, Australia. [5] School of Biology and Environmental Science, Queensland University of Technology, Brisbane, QLD 4001, Australia. [6] Department of Chemical Pathology, Pathology Queensland, Brisbane, QLD 4001, Australia. [7] Faculty of Health and Behavioural Sciences, University of Queensland, Brisbane, QLD 4072, Australia. [8] CSIRO-QUT Synthetic Biology Alliance, Brisbane, QLD 4001, Australia. [9] Centre for Genomics and Personalised Health, Queensland University of Technology, Brisbane, QLD 4001, Australia. [10] Australian Research Council Centre of Excellence for Innovations in Peptide and Protein Science, Melbourne, VIC, Australia. [11] Shemyakin-Ovchinnikov Institute of Bioorganic Chemistry, Russian Academy of Sciences, Moscow 117997, Russia. [12] School of Pharmacy, The University of Queensland, Woolloongabba, QLD 4102, Australia. [13] GeneCology Research Centre, University of the Sunshine Coast, Sippy Downs, QLD 4556, Australia. [14] School of Science, Technology and Engineering, University of the Sunshine Coast, Sippy Downs, QLD 4556, Australia. [15] These authors contributed equally: Yue Wu, Zhenling Cui. ✉email: kirill.alexandrov@qut.edu.au; ziraffa81@gmail.com

Small-molecule drugs and recombinant proteins comprise two main classes of currently used therapeutic agents[1]. While the former is superior in oral bioavailability and the ability to access intracellular targets, they lag behind the latter in potency and selectivity and, as a result, in safety. The demand for novel protein therapeutics for the treatment of cancer as well as chronic and infectious diseases is expected to significantly increase as the global population ages. Yet the developmental risks and the high cost of production remain the major hurdles to the broader use of polypeptide therapeutics[2]. In vitro translation systems that enable rapid prototyping and engineering of recombinant proteins provide an alternative to time-consuming and costly in vivo expression. However, many therapeutically-relevant proteins display complex folding kinetics and rely on co-translational assistance of multiple chaperones and folding catalysts to avoid the formation of kinetically trapped states[3]. Also, an effective coupling of translation and ER-translocation employed by the cell for nearly one-quarter of the proteome ensures sub-cellular segregation of aggregation-prone proteins[4]. Therefore, their cell-free production often results in misfolding and aggregation due to loss of compartmentalization and concerted chaperone activity[5]. To this end, the majority of studies attempt to achieve the productive trade-off between folding and aggregation through the combination of an oxidizing environment for accelerated closure of disulfide bonds with a complex chaperon cocktail[6–8]. However, recent studies[9] suggest that the closure of disulfide bonds does not always direct structure acquisition through the provision of the folding constraints[10], instead, their accelerated formation can often result in randomly cross-linked intermediates[6,7]. Consistent with this, Ryabova et al.[7] showed that bond reshuffling rather than the net formation of disulfide bonds was a prerequisite of efficient folding for some antibody fragments[7]. The effect was observed only co-translationally or shortly following the release of the translated polypeptide chains suggesting a strong aggregation tendency at physiological conditions where folding is partially kinetically controlled. Accordingly, Stech et al.[11], taking advantage of the eukaryotic translation system, showed that optimization of redox conditions was only effective for antibody fraction segregated to the lumen of microsomal vesicles. Intriguingly, a positive influence on the yield of disulfide-rich proteins was observed in E. coli CFS with a mere increase in membrane vesicle surface area[6]. In line with this, a number of studies exploited artificial heterogeneity in CFS through the capturing of translated products either directly to the beads via affinity tags[12] or indirectly via immobilized chaperon component[13], redox component[14], or RNA template[15]. Another study demonstrated the effective refolding of matrix-immobilized proteins[16] that outperformed the chaperone-mediated effect[17].

Peptide-based therapeutics attract increasing interest since they combine pharmacological advantages of small-molecule drugs and protein-based therapeutics. Their rigid, disulfide-stabilized backbones endow them with potency, selectivity, and oral availability[18]. Such modular architecture allows the grafting of heterologous bioactive epitopes into orally deliverable scaffolds[18,19] and enables semi-rational engineering of variants with improved biopharmaceutical properties[20]. Yet, only a small number of unmodified peptide-based drug leads have reached the market due to a lack of efficacy or toxicity concerns at the clinical stage[21].

An alternative strategy in bioactive peptide design is to combine target-recognition and membrane-translocation capability within the same macrocyclic entity[22]. Such a combination of features is difficult to realize without the use of noncanonical modalities and diversity-based screening[22,23]. In the most successful approach, the mRNA library is translated in the fully reconstituted Flexible In vitro Translation (FIT)-system, and unique mRNA-peptide conjugates are further selected on a target[23]. Alternatively, an emulsion bead display, allowing selection of peptides with a broader range of affinities due to the avidity effect of multiple sequence copies[24], can be performed in a crude translation extract and further combined with non-canonical amino acid incorporation using established codon-reassignment techniques[25].

Widely used strategies for peptide prototyping such as chemical synthesis and heterologous expression often result in unsatisfactory yields in relation to time and material costs. Solid-phase synthesis of disulfide-rich peptides either relies on thiol protecting groups to avoid side reactions or demands high initial yields to allow downstream refolding and purification[26]. Also, it can be reliably applied only to peptides shorter than 35[27] or 50[28] residues dependent on beta-sheet proportion. Successful heterologous peptide production generally relies on fusion with a carrier protein to confer solubility[29] and protease protection[30]. This implies downstream processing[28,31] as well as a possible trade-off between carrier cleavage and peptide oxidation[32] often requiring solubilization from inclusion bodies[28]. For both chemical and biological synthetic approaches, it is challenging to produce large amounts of soluble disulfide-rich peptides omitting the refolding step. Robust statistics provided by Venomics platform[27,33], the success rate of peptide refolding following heterologous expression[31] or chemical synthesis[26] suggest that under the universal folding conditions only ~half of the peptides could be produced in a soluble form. This is consistent with the existence of two principal folding modes—"framework" and "collapsed", representing either the hierarchic condensation of native-like elements or the slow flux through the off-path folding states[34], respectively. Therefore, a generally applicable method to accommodate both folding trajectories for cost-effective parallel production of complex disulfide-constrained peptides is expected to facilitate their development into drug-like products[35].

Although over the past decade the cell-free technology made a remarkable progress in the production of complex proteins[36] including full-size antibodies[37] such breakthrough cases requiring the tuning of multiple system parameters remain rather rare. In this study, taking advantage of the universal property of evolutionarily selected polypeptide sequences to acquire native conformation as "all or none" transition in the aggregation-free environment at certain conditions, we suggest a general strategy for prototyping of various disulfide-rich peptides, macrocyclic peptides, and complex disulfide-rich proteins. This approach affords protection of translation products from degradation and aggregation while promoting folding control and enabling the introduction of noncanonical modalities.

## Results

**The cell-free platform for prototyping of complex polypeptides.** We previously reported a solution assay for quantification of in vitro translated peptides carrying a C-terminal RGSIDTWV sequence[38] (RGS peptide). This assay is based on activation of the artificial allosteric protease by the RGS peptide through its interaction with the grafted "affinity clamp" receptor domain of the protease[39] (Fig. 1a). Although it is reliable and quantitative, the assay could not be performed directly in E. coli translation reaction due to proteolytic degradation of the translated peptide as well as reporter peptide (Fig. 1a). This necessitated pretreatment of the translation extract with protease inhibitors and additional removal of endogenous proteases by heat precipitation[38].

We conjectured that capturing the RGS-tagged peptides on "affinity clamp"-coated resin co-translationally would confer protease protection and provide a molecular handle for downstream peptide modification (Fig. 1b). To this end, we prepared "affinity clamp"-coated resin (AC-resin) and determined its ligand-binding

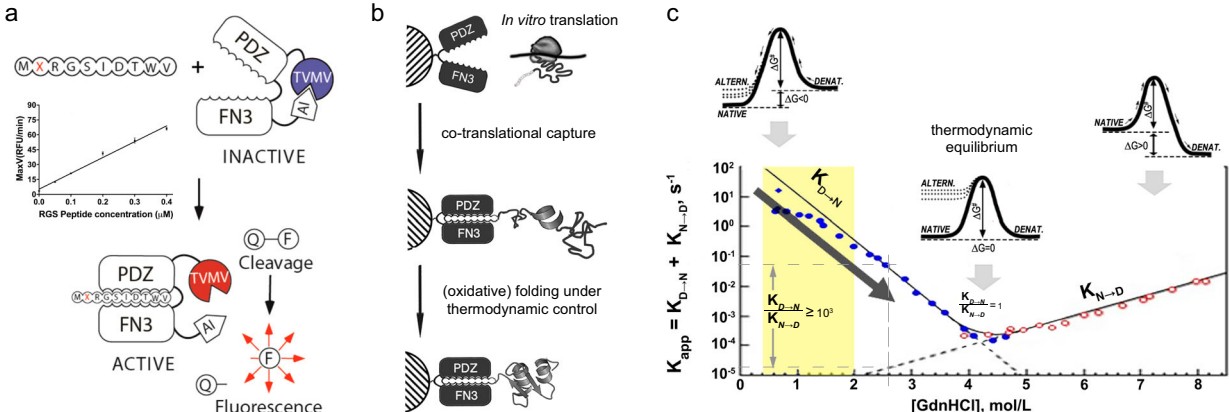

**Fig. 1 Affinity-clamp assay, resin-assisted translation workflow, conceptual overview of protein folding. a** Peptide biosensor consists of an autoinhibited TVMV protease (blue), with PDZ and FN3 domains forming the affinity clamp (AC). Binding of RGSIDTWV-fusion with the peptide of interest (X) to biosensor results in a conformational change of the AC-module that dislodges the autoinhibitory peptide (AI) from the active site of TVMV (red) leading to protease activation and subsequent cleavage of the quenched substrate (Q-F). Q and F denote the fluorescence quencher group and the fluorochrome, respectively. The calibration plot for RGS peptide is obtained by plotting the initial reaction rates against different RGS-peptide concentrations as mean ± s.d. of three independent assays. **b** Workflow for (poly)peptide production in resin-assisted translation reaction; PDZ and FN3 denote the respective capture domains of affinity clamp. **c** Combined kinetic-thermodynamic profile of protein folding. The dependence of folding/unfolding rates on denaturant concentration is illustrated by the chevron plot (modified from Kiefhaber et al.[70]). Blue (filled) and red (open) circles denote the values for folding ($K_{D\rightarrow N}$) and unfolding ($K_{N\rightarrow D}$) rate constants, respectively, at different denaturant concentrations. Schematic Gibbs free-energy diagrams of protein folding correspond to no denaturant (left), its highest concentration (right), and its intermediate concentration, corresponding to a point of thermodynamic equilibrium (middle). ΔG# and ΔG denote folding/unfolding transition state activation energy and the free-energy difference between folded and unfolded states, respectively. NATIVE, DENAT. and ALTERN. denote native, fully denatured, and alternative folding states, respectively. Arrows outlining the free-energy diagrams indicate the shift direction of folding/unfolding equilibrium. The yellow zone highlights the range of guanidine hydrochloride (GdnHCl) concentrations tolerated by affinity-clamp/RGS-peptide complex. The arrow along the left-hand side slope of the plot indicates a shift from kinetically controlled folding space at physiological conditions to thermodynamically controlled folding space upon an increase in denaturant concentration. Upper and lower horizontal dotted lines indicate the corresponding values for folding/unfolding rate constants at arbitrary GdnHCl concentration marked by the vertical line. Numerical ratios of rate constants are displayed at arbitrary concentration and at a point of thermodynamic equilibrium.

capacity, its influence on translation efficiency, and optimal elution conditions (Supplementary Fig. 1A–C). Translation of RGS peptide in AC-resin-supplemented *E. coli* S30-based cell-free system (Ec CFS) containing protease inhibitors resulted in ~20% increase in the yield of RGS peptide compared to standard reaction lacking the affinity resin. However, from eukaryotic *Leishmania tarentolae* extract-based cell-free system (LTE)[40] RGS peptide could be only recovered in the presence of AC-resin despite the pretreatment with protease inhibitors.

Encouraged by these results, we set out to test if the fusion of other peptides to the RGS-tag would equally improve their yield in AC-resin-assisted translation reaction. In order to achieve efficient translation initiation, we used Species Independent Translation initiation Sequence (SITS)[41] comprising an unstructured 5′-UTR, start-codon, and the leader upstream to Peptide of Interest (POI)-coding sequence (Fig. 2a). The coding sequence of tobacco vein mottling virus protease (TVMV)-cleavage site was inserted after the SITS sequence to enable co-translational removal of the leader peptide (Fig. 2a and Supplementary Fig. 1D). We used pLTE- and pOPINE-based plasmid vectors (Supplementary Table 3) to evaluate the translation efficiencies for several such assemblies under control of the SITS or a classical Shine-Dalgarno motif, respectively. The peptide assemblies included RGS fusions with the smallest (SFTI) or largest (Dc1a) peptide-coding ORFs used in this study (Table 1). As a result, SITS consistently supported the highest peptide yields in resin-assisted Ec CFS as well as in a fully reconstituted PURE system (Fig. 2b and Supplementary Fig. 2A). Expectedly, the full SITS outperformed the rest of the tested initiation sequences in the eukaryotic LTE system (Supplementary Fig. 2C), however overall productivity of LTE compared to Ec CFS was ~tenfold lower for peptide translation (Supplementary Fig. 2D). Similar product

yields obtained for different ORF arrangements in Ec CFS and PURE system, despite more than twofold ribosomal enrichment in the latter (Supplementary Fig. 2B), indicated that the productivity of the former was not limited by compromised stability of peptide-coding RNA (Supplementary note 1). In agreement with the previous report[42], we found near-linear inverse relationship between (poly)peptide's molar yield and size in the range from 49 to 315 amino acids (Supplementary Table 1) suggesting the inactivation of some key translation component(s) or resource(s) rather than inefficient ribosome recycling to be the major factor limiting the productivity of resin-assisted Ec CFS (Supplementary Table 1, Supplementary Notes 2 and 3 and Supplementary Fig. 3A, B). Accordingly, translation in continuous exchange format resulted in ~4–5-fold higher peptide yield (Supplementary Fig. 4). We also found that AC-resin can be regenerated up to seven times by denaturing/refolding procedure.

**Peptide on-resin reduction and oxidative folding.** The ability of AC-coated resin to withstand denaturants and reducing agents without loss of ligand-binding activity encouraged us to test it as a solid-phase handle for peptide folding and cyclization. As test examples, we chose cyclic peptides of varying complexity such as the sunflower trypsin inhibitor (SFTI) comprising a single disulfide bond which has been used as a versatile template for the engineering of protease inhibitors[43], and the cyclotide MCoTI-II from the melon plant *Momordica cochinchinensis,* featuring a complex cystine knot[18] (Fig. 3a, f).

RGS fusion with circularly permuted SFTI (prmSFTI-RGS) opened at the scissile bond between Lys-5 and Ser-6 (Fig. 3a) could be eluted from the resin with DMSO following translation in resin-assisted Ec CFS. Subsequent treatment of the peptide with an

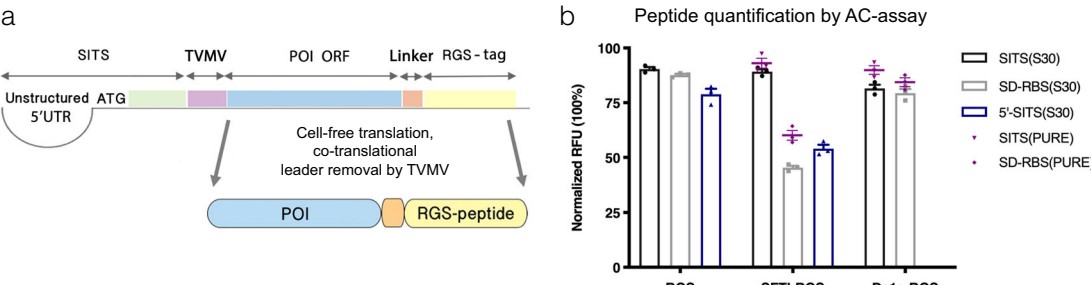

**Fig. 2 Design and evaluation of expression cassette. a** Design of expression construct: universal initiation sequence (SITS) comprising the unstructured 5′ UTR, start-codon (ATG) and 3′ part, encoding for the leader peptide (green), directs translation of peptide-of-interest (POI) carrying the RGS-tag (yellow). A TVMV-cleavage site (pink) is included for co-translational removal of the SITS-derived translation leader. **b** Comparative performance of different translation initiation sequences including full SITS, classical Shine-Dalgarno Ribosome Binding Site (SD-RBS) and unstructured 5′UTR derived from SITS (5′-SITS) in supporting the synthesis of peptides from different size classes such as 8 aa RGS peptide (RGS), RGS fusion with 14 aa SFTI (SFTI-RGS) and 57 aa Dc1a (Dc1a-RGS) (Table 1). Peptides were quantified using affinity-clamp assay and initial rates of substrate cleavage are plotted as bars or dots for peptide variants produced in S30-based (S30) or PURE (Protein synthesis Using Recombinant Elements) translation systems, respectively. The results are represented as means ± s.d. of $n = 3$ repeats of the assay.

equimolar amount of immobilized trypsin resulted in excision of linear prmSFTI followed by its cyclization resulting in fourfold excess of cyclic SFTI over its linear permuted form (Supplementary Fig. 5A and Supplementary Note 4) while tenfold excess of the peptide over trypsin mostly yielded the linear prmSFTI accordingly to "Laskowski mechanism" (Supplementary Fig. 5B and Supplementary Note 4). Cell-free produced linear, oxidized prmSFTI displayed inhibition potency similar to that of its chemically synthesized linear counterpart (Fig. 3c and Supplementary Fig. 6A, B). However, the linear prmSFTI possessed slightly weaker trypsin-inhibitory activity than the cyclic form (Supplementary Fig. 6A and Supplementary Note 4) due to secondary cleavage at Arg-2 resulting in the formation of doubly cleaved product (Fig. 3b and Supplementary Fig. 5B) in agreement with the previous report[40].

Synthesis of wild-type MCoTI-II (wtMCoTI) and its circularly permuted version (prmMCoTI) opened at the scissile bond between Lys-6 and Ile-7 and fused to RGS in resin-assisted Ec CFS led to poor recovery of both translation products from AC-resin (Fig. 3e and Supplementary Fig. 7A) similar to previously observed poor elution of cysteine-rich polymeric SFTI (Supplementary Fig. 3B). Most likely this was a consequence of peptide aggregation via disulfide bond scrambling—a common phenomenon for cysteine-rich peptides[44] that results from kinetic domination of covalent interactions over non-covalent interactions[34]. Following incubation of the resin-bound MCoTI with 50 mM DTT, both MCoTI-RGS variants could be recovered in a fully reduced and soluble state (Fig. 3d, e and Supplementary Fig. 8A). Various degrees of aggregation were observed for other peptides with two and three disulfide bonds demanding full reduction to achieve 100% elution efficiency (Supplementary Fig. 7B). In the next step, immobilized and reduced peptides were subjected to on-resin oxidation in the presence of reduced glutathione (GSH) to allow disulfide-bond reshuffling and equilibration into the most stable native fold under thermodynamic control (Supplementary Note 5). Being "two-state" folders[45], peptides are likely to possess similar dependence of the folding kinetics on reductant concentration as displayed by the proteins relative to denaturant concentration (Fig. 1c). Similarly, thermodynamically controlled folding at increased reductant concentration would proceed slower due to a decrease of the folding rate constant. After varying the reductant concentration and incubation time, we resorted to ~12–48 h incubation with 10 mM GSH to allow complete recycling of kinetically trapped "off-pathway" states. For both MCoTI variants, this protocol yielded dominant products with three disulfide bridges (Fig. 3f

and Supplementary Fig. 8B). While prmMCoTI had a trypsin inhibition constant similar to that measured previously for the intact cyclic McoTI[46], confirming the formation of the native cystine knot (Supplementary Fig. 6C), wtMCoTI-RGS displayed almost fourfold higher Ki for trypsin due to increased flexibility in the open backbone (Supplementary Fig. 6C). We conjectured that subjecting fully reduced and spatially segregated peptides to oxidative folding under thermodynamic control can serve as a generic approach for the formation of complex peptide folds.

**Prototyping of diverse disulfide-constrained peptides.** Next, we explored the general applicability of the established platform for the production of peptides of biomedical and commercial value featuring diverse lengths and topologies. Besides SFTI and MCoTI our test set included 23 aa arenicin-3 analog AA139 that shows antimicrobial activity, Kalata B1 - another plant cyclotide that serves as design framework for grafting and delivery of various biological activities, two highly selective sodium channel inhibitors Pn3a and Dc1a as well as Holocyclotoxin-1 from paralysis tick *Ixodes holocyclus* (Table 1, Supplementary Figs. 9–12, and Supplementary Notes 6–9). Small-size peptides were effectively amenable to chemical synthesis as described previously. Recombinant[35] or chemical synthetic routes were previously reported for McoTI and kalata B1[34]. However, the rest, partially undergoing "collapsed" folding[34], demanded a downstream refolding at optimized redox conditions despite their expression as fusion with MBP in *E. coli* periplasm or in the engineered *E. coli* strain promoting oxidative folding[47,48]. Holocyclotoxin-1 (HT-1) could be previously assembled by a combination of SPPS and native chemical ligation[49]. Its heterologous production in the prokaryotic system was unsuccessful while its fusion with the protein carrier demanded an additional refolding step following production in yeast[50]. We translated these peptides in resin-assisted Ec CFS and, following co-translational matrix capture and full on-resin reduction, confirmed purity and integrity of Kalata B1- (Supplementary Fig. 8C, 9A), Pn3A- (Supplementary Fig. 12A) and SFTI-RGS fusions (Supplementary Fig. 13A) by MALDI-MS. We further confirmed the formation of homogeneous oxidation products following on-resin oxidative folding at established conditions (Table 1) for MCoTI, Kalata B1 (Supplementary Fig. 8B, D), AA139 (Supplementary Fig. 10B), HT-1 (Supplementary Fig. 11B), Pn3a, and Dc1a (Supplementary Fig. 12D, F). To produce a native cyclic Kalata B1, the oxidation was performed following backbone

**Table 1 Summary of disulfide-constrained peptides produced in AC-assisted Ec CFS.**

| Name | Schematic structural view of final peptide variants assayed | [a]N/C | Spacer | [b]GSH/ time | [c]RGS- removal | [d]Assay | POI-RGS [e](µM) | Figure | [f]SN |
|---|---|---|---|---|---|---|---|---|---|
| SFTI | SIPPICFPDGRCTK | prm | GAG | DMSO/NA | Trypsin | TIA | 9.5 | 3a | 4 |
| McoTI | ILKKCRRDSDCPGACICRGNGYCGSGSDGGVCPK | prm | GAG | 10 mM/12 h | Trypsin | TIA | 6.4 | 3f | 5 |
| Kalata B1 | GLPVCGETCVGGTCNTPGCTCSWPVCTRN | wt | GL | 5 mM/ 12 h | AEP | Co-elution | 8.0 | S9 | 6 |
| AA139 | GFCWVYCARRNGARVCYRRCN | wt | GL | 10 mM/12 h | NH2OH | Micr | 4.9 | S10 | 7 |
| HT-1 | SCTNPGKKRCNAKCSTHCDCKDGPTHNFGAGPVQCKKCTYQFKGEAYCKQ GAG RGSIDTWV | wt | GAG | 10 mM/48 h | NA | DHFR | 7.6 | S11 | 8 |
| Pn3a | DCRYMFGDCEKDEDCCKHLGCKRKMKYCAWDF | wt | NGLP | 10 mM/12 h | Carb.Y | Co-elution | 5.0 | S12 | 9 |
| Dc1a | SAKDGDVEGPAGCKKYDVECDSGECCQKQYLWYKWRPLDCRCLKSGFFSSKCVCRDV GTGSGG R | wt | GTGSGG | 10 mM/48 h | Th. | Fly | 8.9 | S12 | 9 |

Disulfide connectivity and backbone cyclization (if applicable) are shown above and below the sequences, respectively; [a]wt or prm denote wild type and circularly permuted N-/C-termini arrangement in the linear RGS-tagged peptide precursor; [b]oxidative folding conditions are indicated as glutathione (GSH) concentration/incubation time; [c]RGS-tag removal with asparagine endopeptidase (AEP), – hydroxylamine (NH2OH), carboxypeptidase Y (Carb. Y), thrombin (Th.); [d]assay used for functionality test: trypsin-inhibitory assay (TIA), antimicrobial assay (Micr), assay based on interference with activation of dihydrofolate reductase (DHFR) (Supplementary Note 8), co-elution with folded peptide controls, insecticidal assay (Fly); [e]µM yield of POI-RGS in resin-assisted translation reaction estimated by affinity-clamp assay; [f]reference to respective Supplementary Note.

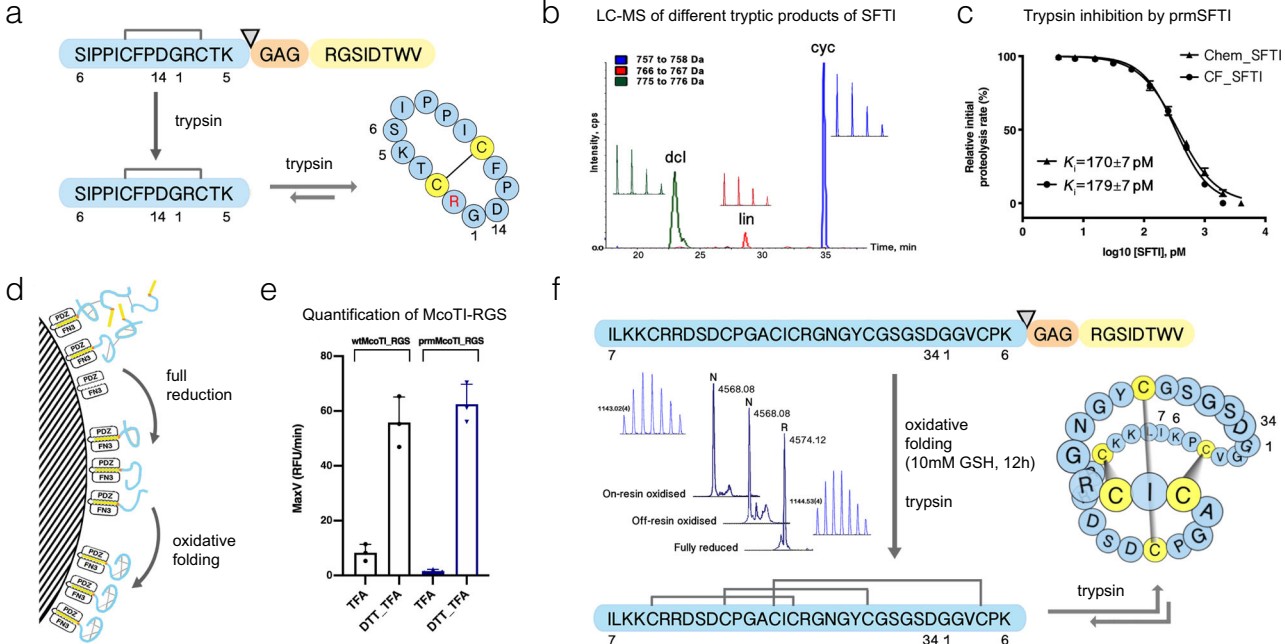

**Fig. 3 On-resin modification of model peptides produced in AC-resin-assisted Ec CFS. a** Trypsin-mediated RGS-tag removal from SFTI-RGS fusion and cyclization of circulalry permuted SFTI. **b** LC-MS profiling of SFTI backbone variants following trypsin treatment; dcl, double-cleaved (at Arg-2, Lys-5); lin, linear circularly permuted SFTI; cyc, backbone-cyclized SFTI. Insets indicate the respective monoisotopic mass peaks. **c** Comparison of trypsin-inhibitory activities of chemically (chem) or cell-free (CF) synthesized linear oxidized circularly permuted SFTI (prmSFTI). The graph represents the results as means ± s.d. of $n = 3$ repeats of the assay. $K_i$ was determined by $IC_{50}$ using a tight-binding equation (see Methods). **d** Schematics of on-resin peptide modification steps. Representative peptides are color-coded as in (**a**). **e** AC-assay quantification of wild-type (wtMcoTI) and circularly permuted (prmMcoTI) MCoTI-RGS variants eluted from resin with 0.2% trifluoroacetic acid before (TFA) or after full reduction with 50 mM DTT (DTT_TFA), respectively. The initial rates (RFU/min) of substrate cleavage are plotted as means ± s.d. of $n = 3$ independent assays. **f** Oxidative folding of circularly permuted MCoTI followed by trypsin-mediated RGS-tag removal and cyclization. Dominant LC/MS peaks correspond to either fully reduced or oxidized form of MCoTI-RGS. Cystine knot installed following a closure of trypsin-inhibitory loop of 11-membered embedded ring is shown schematically and participating cysteine residues are highlighted in yellow. Insets indicate the respective monoisotopic mass peaks.

cyclization since the fully reduced peptide was reported[51] to be a better substrate for cyclization by asparagine endopeptidase. Hence, we followed the originally published protocol (Supplementary Note 6).

Production of these peptides in a functional state also requires the removal of the RGS-tag. In the case of kalata B1, this was achieved using asparagine endopeptidase (Supplementary Fig. 9B, C). However, this approach was unsuccessful for AA139 which was processed by hydroxylamine-mediated cleavage (Supplementary Fig. 10). We then tested the ability of exopeptidases to remove the RGS-tag using amino acid "stopper". Indeed, immobilized carboxypeptidase A removed the RGS-tag nearly quantitatively downstream the Arg-stopper of RGS peptide (Supplementary Fig. 13B and Supplementary Note 10). In an alternative approach, thrombin efficiently cleaved the Arg-Gly bond in the RGS motif without affecting alternative cleavage site(s) within the constrained core of disulfide-bonded peptide (Supplementary Fig. 13C and Supplementary Note 10). We propose that these two approaches for the removal of the RGS-tag would enable the preparation of near-native forms for the majority of peptides. Alternatively, the application of Strep-Tactin affinity resin supporting immobilization through the N-terminal tag would enable seamless tag removal using Tobacco Etch Virus (TEV) or TVMV proteases. The generated AA139 and Dc1a peptides showed the expected biological activity in antimicrobial and insecticidal assays, respectively (Supplementary Figs. 10C and 12G and Supplementary Notes 7 and 9). HT-1 functionality was confirmed by its ability to interfere with DHFR-calmodulin-sensor activation (Supplementary Fig. 11C and Supplementary Note 8). The native structure of kalata B1 and Pn3a were evidenced by co-elution with their correctly-folded synthetic versions (Supplementary Figs. 9C and 12D and Supplementary Notes 6 and 9).

**Production of the peptide with non-native intramolecular bond.** We previously reported selective and efficient inactivation of endogenous *E. coli* tRNA$^{Ser}$GCU and tRNA$^{Arg}$CCU directly in the S30 cell extract using modified antisense oligonucleotides. This approach liberates AGT and AGG codons respectively and enables their reassignment to non-canonical amino acids (ncAAs)[25]. To test if the approach we developed for in vitro translation and capture of peptides could be applied for generation of peptide harboring ncAAs, we set out to replace Cys3 and Cys11 of SFTI with *n*-propargyl-L-lysine (Prk) and *p*-azido-L-phenylalanine (AzF) (Fig. 4a and Supplementary Note 11). As can be seen in Fig. 4, we were able to simultaneously reassign chosen codons to Prk and AzF and perform on-bead peptide cyclization via copper-catalyzed click reaction. We could only detect the peptide product comprising two click-chemistry labels from the resin-assisted reaction while no product could be captured post-translationally following the standard translation reaction. The formation of an internal 1,4-disubstituted 1,2,3-triazole bond was confirmed by LC-MS analysis of purified peptides before and after the click reaction (Fig. 4b, c). This demonstrates the possibility of generating macro-cyclized peptides with non-native bonds using a combination of AC-assisted in vitro translation in a crude Ec CFS with the simultaneous reassignment of two sense-codons.

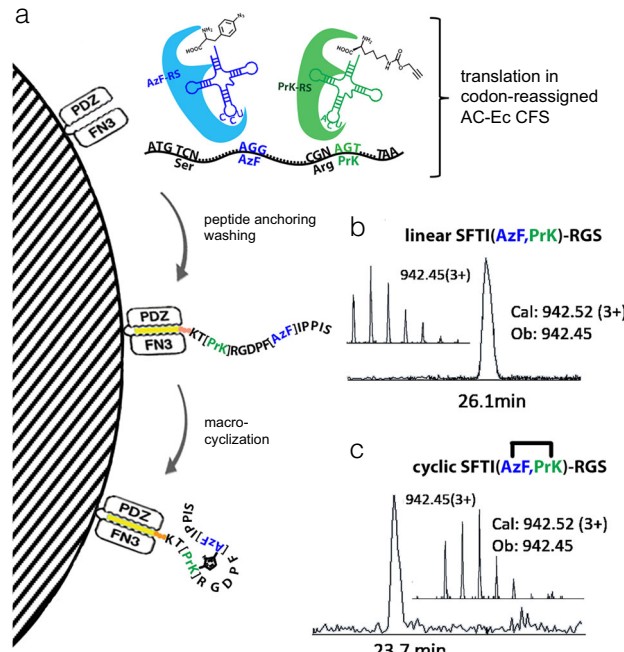

**Fig. 4 Production of the macrocyclic peptide with non-native bond in codon-reassigned AC-resin-assisted Ec CFS. a** Peptide production steps: translation of linear peptide in AC-resin-assisted Ec CFS (top) using codon-biased template pLTE-SFTI(agg,agt)-RGS where the unique AGG and AGT codons were reassigned to p-azido-L-phenylalanine (AzF) and n-propargyl-L-lysine (PrK) via inactivation of their cognate native tRNAs while their synonymous CGN and TCN codons were reserved to continue decoding Arg and Ser. Orthogonal synthetase/tRNA pairs are shown in blue and green; peptide co-translational capture and washing of resin-immobilized peptide (middle) followed by on-bead click-chemistry mediated formation of triazole bridge (bottom). **b, c** LC-MS elution profile corresponding to linear (**b**) and macrocyclic (**c**) SFTI-RGS fusions, respectively. Insets indicate the same monoisotopic masses due to no mass change following cycloaddition.

**Co-translational resin capture reduces protein aggregation.** Bacterial translation surpasses eukaryotic translation in productivity owing to a less strict translational control. However, incompatibility of folding machineries leads to low folding yields for complex multidomain eukaryotic proteins[5,52]. Furthermore, in cell-free translation systems, cooperation of ribosome-associated and cytosolic chaperones is likely to be partially impaired, leading to exposure of semi-folded multidomain proteins to each other and to crowded environments[53]. We first conjectured that the inclusion of the affinity resin into the reaction mixture would help to avoid the common trade-off between folding and aggregation through the sequestration of on- and off-path folding intermediates to the bead surface (Fig. 1b). We assumed that the use of biological interaction of exquisite selectivity and affinity would serve to avoid contamination with endogenous proteins often associated with immobilized-metal-affinity chromatography[42,54]. We further assumed that such spatial segregation would provide an aggregation-free environment for matrix-assisted recycling of folding intermediates under thermodynamic control (Fig. 1b). For comparative estimation of protein solubility and activity in response to the addition of affinity matrix, we used eGFP fusions with dihydrofolate reductases derived from E. coli (eDHFR) or human (hDHFR), thereby allowing us to independently monitor the folding states of both fusion partners. The choice of eDHFR and hDHFR was based on differences in their folding efficiencies, where eDHFR was previously shown to fold rapidly and efficiently in a variety of fusion

contexts. In contrast, hDHFR was shown to undergo spontaneous aggregation and misfolding[55]. These folding reporters carrying both N-terminal Twin-strep-tag and C-terminal RGS-tag were translated in both E. coli-based (Ec CFS) and Leishmania-based (LTE) cell-free transcription–translation systems in the presence of either Strep-Tactin- or AC-coated resins. Flow-through from each reaction was separated into soluble and insoluble fractions while Strep-Tactin-resin-bound protein was eluted with the biotin-containing buffer. The immunoblot analysis revealed similar total yields for translated proteins regardless of co- or post-translational resin supplementation (Fig. 5a, panel 1 and Supplementary Fig. 14). However, large insoluble fraction was detected in the latter case (Fig. 5a, panel 2 and Supplementary Fig. 14). Comparative fluorescence analysis of protein fractions revealed ~2–3-fold more fluorescence associated with the Strep-Tactin-coated resin from the resin-assisted reaction compared to protein fractions captured post-translationally (Figs. 5b and 6a). Less apparent difference between co- and post-translationally associated protein fractions was observed for AC-resin-assisted reaction format that requires the exposure of C-terminal RGS-tag thus precluding capture before the full release of the translated product (Supplementary Fig. 15A). Resin-assisted reactions supplemented with biotin failed to improve the yield of fluorescent protein fraction (Fig. 5b) resulting in a similar amount of insoluble material as in the standard translation reaction lacking the resin (Fig. 5a, panel 2). This indicates that the gain in fluorescence could be specifically attributed to resin-mediated interference with protein aggregation. Interestingly, comparative analysis of co- and post-translational reaction formats revealed that in the latter case all studied proteins formed insoluble aggregates which could no longer associate with the resin post-translationally while the captured fractions most likely corresponded to the protein that succeeded to avoid the aggregation and fold correctly during the reaction time course (Fig. 5a, panel 3). The disproportion between the total co- and post-translationally captured protein fractions (Fig. 5a, panel 3) and their respective fluorescent yields (Fig. 5b) suggests that all co-translationally captured fractions included from ~25 to ~50% of likely misfolded or semi-aggregated nonfluorescent material. Fluorescent fractions of GFP and GFP-eDHFR could be eluted near completely from the resin while only 70% fluorescence could be recovered following the elution of co-translationally captured GFP-hDHFR (Fig. 5b). The addition of 0.125% Tween 20 into the elution buffer increased the fluorescence recovery (Fig. 5d), suggesting that the insoluble fraction remaining on the resin was a likely result of a delayed aggregation via misfolded hDHFR part. We next performed the DHFR-assay on eluted fractions to roughly estimate whether both fusion partners were folded to a similar extent.

For GFP-eDHFR, a direct correlation between GFP fluorescence level and DHFR activity in co- and post-translationally captured samples was observed, suggesting that both fusion partners were folded to a similar degree (compare Fig. 5b, c for co-/post-). Surprisingly, when expressed in Ec CFS but not in LTE, GFP-hDHFR displayed an inverse correlation between fluorescence level and DHFR activity with larger proportion of active hDHFR in the fraction captured post-translationally but not co-translationally in contrast to the respective fluorescence yields (Fig. 5b, c). However, direct correlation between fluorescence level and hDHFR activity in co- and post-translationally captured fractions was observed in LTE (Fig. 6a, b). This phenomenon indicates that exposure of hDHFR fusion to some component in the bacterial lysate, in alternative to its immobilization to the solid phase, increases the proportion of active hDHFR (see below). This is consistent with the predominance of post-translational component over co-translational in protein folding in the prokaryotic system in contrast to eukaryotic system[5,52]. Accordingly, a co-translational folding, typical to eukaryotic translation systems,

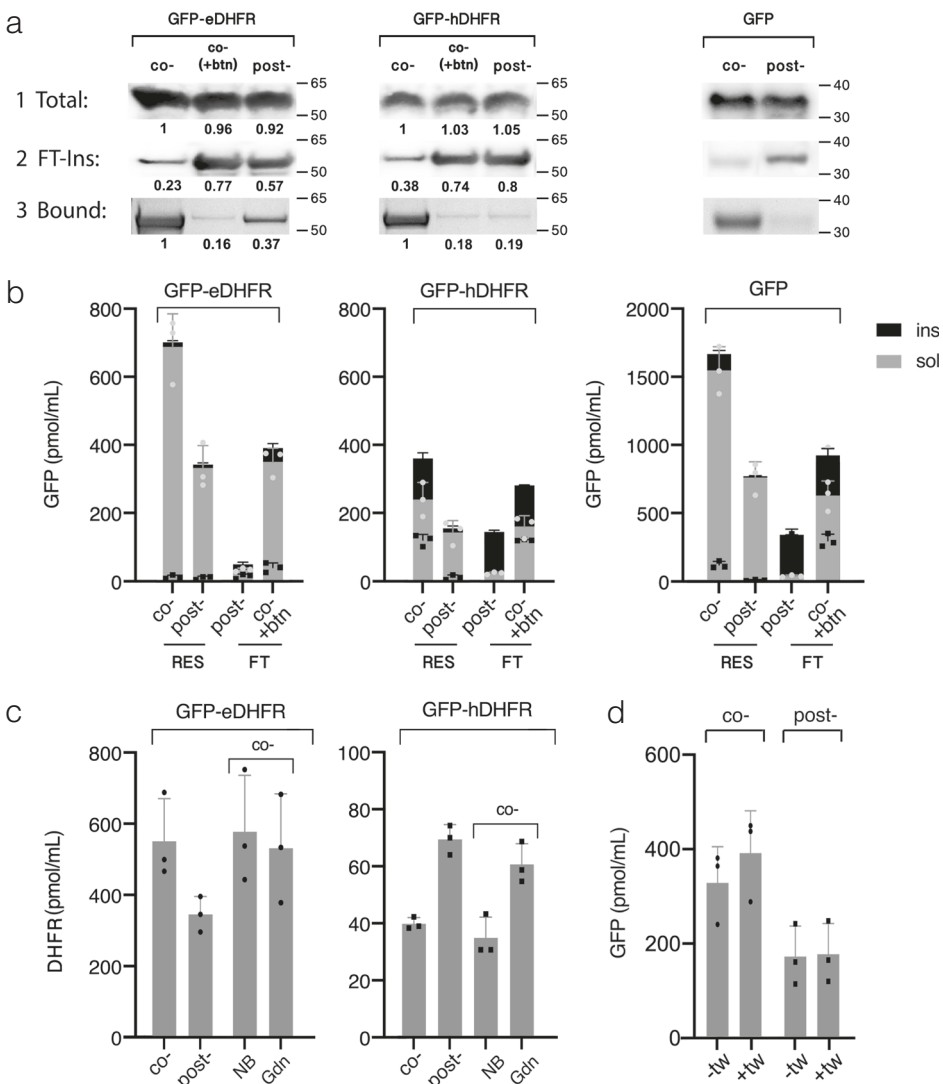

**Fig. 5 Analysis of folding reporters translated in Strep-Tactin-resin-assisted Ec CFS. a** Immunoblot of protein bands derived from unfractionated translation reactions (1 Total) or insoluble fraction from flow-through (2 FT-ins) probed with anti-GFP antibodies. Panel 3 shows Coomassie-stained resin-bound protein fractions (3 Bound), eluted and resolved on SDS-PAGE. Co- and post- denote the respective reaction formats, co-(+btn) contains 10 mM biotin to prevent protein capture onto resin. The band densities were obtained as non-saturated integrations of the respective image bands using ImageJ software and normalized either to "Total, co-" for panels 1, 2 or to "Bound, co-" for panel 3. **b** Fluorescence analysis of resin-bound (RES) and free-unbound (FT) fractions following co- and post-translational protein capture. Gray defines soluble fractions either eluted with biotin from the resin or remaining in the flow-through after centrifugation at 20 kg for 30 min at 4 °C. Black defines insoluble fractions remaining on resin or removed from flow-through following elution or centrifugation, respectively. Fluorescence of insoluble fraction does not account for misfolded protein and only provides the relative estimation of the degree of protein aggregation. Fluorescence units were converted to picomoles of GFP using the respective calibration curves (Supplementary Fig. 16A, C) and further adjusted to 1 ml of translation reaction (for FT) or to resin amount used for 1 ml of translation reaction (for RES). Results are plotted as means ± s.d. of $n = 3$ independent experiments. **c** DHFR activity analysis in eluted protein fractions following co- or post-translational capture before (co-/post-) or after (NB/Gdn) on-resin treatment of co-translationally captured protein with 1.6 M guanidine hydrochloride-containing buffer (Gdn) or with neutral buffer lacking denaturant (NB) for 2 h at room temperature. Initial rates of fluorescence change for NADPH-oxidation reactions were converted to picomoles of active DHFR using the respective calibration plot (Supplementary Fig. 16B) and further adjusted as in (**b**). The results are plotted as means ± s.d. of $n = 3$ independent experiments. **d** Fluorescence analysis of GFP-hDHFR fractions eluted with standard elution buffer (-tw) or with the buffer containing 0.125% Tween 20 (+tw). The results are represented as means ± s.d. of $n = 3$ independent experiments.

may explain the observed direct correlation between DHFR and GFP levels as well as similar fluorescence yields for fast-folding GFP and GFP-eDHFR regardless of co- or post-translational reaction formats in LTE (Fig. 6a).

**Recycling of misfolded states under thermodynamic control.** In both translation systems for GFP-hDHFR fusion, the molar yield of fluorescent GFP exceeded the yield of active hDHFR. This suggests several folds to order of magnitude excess of misfolded hDHFR fraction over its active fraction for different reaction formats as can be seen from the data compilation shown in Supplementary Table 2.

The physical model of protein folding predicts that under physiological conditions single-domain proteins above 100 amino acid residues in size fold under various degrees of kinetic control[56]. To bypass the kinetic traps and complete the folding in a biologically reasonable time at physiological conditions in vivo,

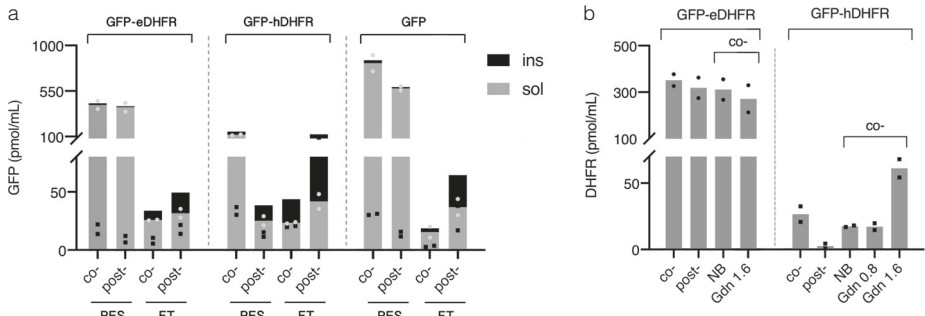

**Fig. 6 Analysis of folding reporters translated in Strep-Tactin-resin-assisted *Leishmania*-based cell-free system (LTE).** **a** Fluorescence analysis of resin-bound (RES) and free-unbound (FT) protein fractions following o- and post-translational protein capture (co-/post-). Gray and black define soluble and insoluble fractions, respectively, as described in Fig. 5b. Fluorescence units were converted to picomoles of GFP obtained from 1 ml of translation reaction as described in Fig. 5b. The results are plotted as means of two independent translation experiments. **b** DHFR activity analysis in eluted protein fractions following co- or post-translational capture before (co-/post-) or after the treatment of co-translationally captured protein with the buffer containing either no (NB), or 0.8 M (Gnd 0.8), or 1.6 M (Gnd 1.6) of guanidine hydrochloride as described in Fig. 5c. Initial rates of fluorescence change of NADPH-oxidation reactions were converted to pmol/ml of active DHFR as described in Fig. 5c. The results are plotted as means of two independent translation experiments. Source data are provided as a Source Data file.

chaperones are often required to both stabilize high-energy folding intermediates and protect them from aggregation. This is in particular relevant to high contact order proteins such as DHFR stabilized by long-range interactions which either possess fast-folding kinetics or rely on concerted chaperon assistance[57]. In cell-free systems, the latter can be partially impaired and its reconstitution requires optimization of multiple system parameters[6–8]. However, evolutionarily selected polypeptides, sometimes viewed as "aperiodic crystals", behave as two-state folders at domain level upon approaching the thermodynamic equilibrium[58]. We conjectured that treatment of immobilized protein with denaturant in a range of concentrations within a distance from the point of thermodynamic equilibrium would allow slow recycling of misfolded states under thermodynamic control (Fig. 1c). Since misfolded states by definition are less stable than the native state their unfolding would progressively increase upon approaching the thermodynamic equilibrium along the left-hand side slope of the chevron plot (Fig. 1c). At the same time, along this slope the rate of protein folding exceeds by several orders of magnitude the rate of native structure unfolding with the latter remaining a rare "all or none" process (Fig. 1c). Therefore, we assumed that misfolded states can be recycled to the native state on-resin while maintaining the complex between protein and affinity module. Its maintenance is important since proteins in semi-denatured state are prone to aggregation due to solvent-exposed hydrophobic groups[59]. We tested its stability with recombinantly purified GFP carrying both peptide tags and demonstrated that >80% of purified GFP retains on Affinity-Clamp- and Strep-Tactin-cotated resins following 6 h and 2 h incubation at RT in the buffers containing 2 M and 1.6 M of guanidine hydrochloride (GdnHCl), respectively (Supplementary Figs. 15C and 17A). Following treatment with denaturant, we observed only minor improvement in eDHFR-activity for co-translationally captured GFP-eDHFR in both translation systems consistent with its partially folded initial state (Fig. 5c, NB/Gdn). However, a 3.5-fold increase in hDHFR activity of GFP-hDHFR produced in resin-assisted LTE was observed following treatment with GdnHCl at 1.6 M but not at 0.8 M (Fig. 6b, NB/Gdn and Supplementary Fig. 18B). Interestingly, only a minor or no increase in the yield of fluorescent GFP following denaturant treatment suggested that the observed gain in DHFR activity mostly stemmed from the recycling of intrinsically misfolded hDHFR or its intermolecular aggregates rather than from

dissolving its intramolecular aggregate with GFP (Supplementary Fig. 17B, C). In contrast to LTE translation, the activity of hDHFR-fusions produced in resin-assisted Ec CFS varied upon denaturant treatment in a batch-dependent manner ranging from ~2.5-fold improvement for the "reactive" batch (Fig. 5b, NB/Gdn and Supplementary Fig. 18A) to no improvement for the "silent" batch following 2 h treatment with 1.6 M GdnHCl buffer. Interestingly, an increase in hDHFR activity, partially masked by protein dissociation, could be obtained for the "silent" batch following extended 12 h treatment of AC-captured GFP-hDHFR with 2 M GdnHCl-containing buffer (Supplementary Fig. 15B). This discrepancy, likely stemming from the variability in chaperon retention between different lysate batches, points at some structural feature responsible for the slow step in hDHFR (re)folding following translation in Ec CFS production but not in LTE. We surmised that the cis-Pro66 in hDHFR, absent in eDHFR, might be such a structural element since proline isomerization is a well-known rate-limiting step for protein folding in vitro. The requirement of chaperone assistance for the peptidyl-prolyl isomerase (PPIase)-dependent catalysis of unfavorable trans-to-cis isomerization in vivo[60] can provide a rationality for the prolonged protein exposure to lysate in order to gain activity in Ec CFS but not in LTE where co-translational chaperone assistance is likely to facilitate the formation of cis-Pro motif by significant protein fraction co-translationally. Indeed, the replacement of Pro66 with Ala in GFP-hDHFR(P66A) restored the correlation between DHFR activity and GFP fluorescence in Ec CFS (Fig. 7a, b) and eliminated the slow step in its refolding compared to a control lacking this mutation (Fig. 7b, NB/Gdn). Conversely, we found that rapamycin but not cyclosporine A eliminated the beneficial effect of protein exposure to lysate in Ec CFS suggesting FKBP-like PPIase was responsible for post-translational catalysis (Supplementary Fig. 19). However, for the majority of proteins from PDB containing cis-prolyl bonds, folding constraints seem to largely facilitate the isomerization reaction[61], whereas hDHFR may represent an exceptional case appearing to be one of the least soluble proteins as reported by Hirano et al.[55] based on analysis of the large combinatorial protein subset. Therefore, prolonged treatment of isomerization-challenged proteins immobilized onto AC or metal-affinity resin withstanding higher denaturant concentrations is likely to favor a gain of correct bond configuration by larger protein fraction.

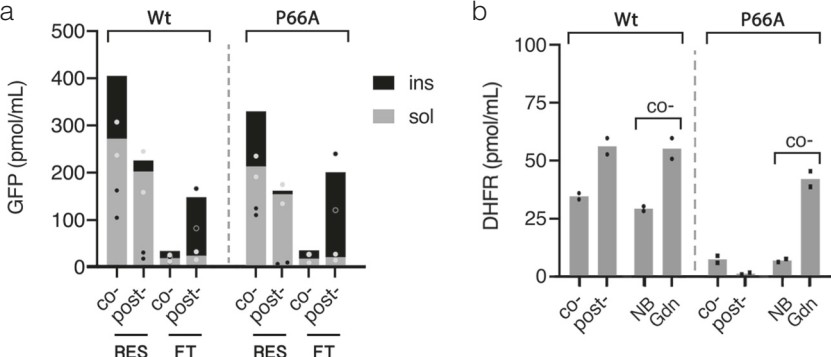

**Fig. 7 Effect of Pro66 to Ala replacement in hDHFR on GFP-hDHFR activity following translation in Strep-Tactin-resin-assisted Ec CFS. a** Fluorescence analysis of resin-bound (RES) and free-unbound (FT) fractions of wild-type (wt) and Pro66Ala mutant (P66A) of hDHFR in GFP-hDHFR following co- and post-translational protein capture (co-/post-). Gray and black define soluble and insoluble fluorescence proportions, respectively, as described in Fig. 5b. Fluorescence units were converted to picomoles of GFP obtained from 1 ml of translation reaction as described in Fig. 5b. The results are plotted as means of two independent translation experiments. **b** Comparative analysis of mutant (P66A) and wild-type (wt) hDHFR in the context of GFP-hDHFR fusion protein eluted following co- or post-translational capture before (co-/post-) or after on-resin treatment of co-translationally captured protein with the buffer containing either no (NB), or 1.6 M (Gnd) of guanidine hydrochloride as described in Fig. 5c. Initial rates of fluorescence change of the respective NADPH-oxidation reactions were converted to pmol/ml of active DHFR as described in Fig. 5c. The results are plotted as means of two independent translation experiments. Source data are provided as a Source Data file.

**Prototyping and analysis of disulfide-constrained proteins**. The trade-off between disulfide-bond closure and protein aggregation[7] is often required for cell-free production of disulfide-constrained antibody fragments[6]. By separating protein translation and folding to different reaction compartments, the proposed approach eliminates this trade-off allowing independent optimization of these two processes. To this end, we took advantage of the tolerance of Ec CFS to high reductant concentrations and translated various antibody fragments in the presence of 10 mM DTT, in diluted lysate at optimal temperature (Supplementary Fig. 20A–C), taking into account that the frequency of intermolecular collisions is proportional to the square of protein concentration and the root square of temperature[62]. Using these conditions in resin-assisted Ec CFS, we demonstrated a marked improvement in the production of various antibody fragments as compared to a normal batch reaction (Fig. 8a and Supplementary Figs. 20C, D, 21). By pull-down assay with the respective ligands, we demonstrated that a fraction of functional Fab, scFv, and VHH antibody fragments was further increased following treatment with denaturant and reductant. We also proved that the pulled ligand was derived from the antibody–ligand complex by cleaving it off the resin by Prescission protease (Fig. 8b). Next, taking advantage of SITS we conducted a comparative analysis of resin-assisted Ec CFS and LTE performances versus widely adopted HeLa cell-free transcription–translation system[63] toward the production of antibody fragments and receptor-binding domain of SARS-CoV-2 (CoV2-RBD). In contrast to our previous benchmarking assay[64], where we compared protein integrities and aggregation propensities between different systems in regular reaction setups, in the current study, we analyzed the changes in folded fractions. Ec CFS was found to surpass LTE and HeLa in eGFP expression level by three- and fourfold, respectively (Supplementary Fig. 22), therefore protein samples from 20 μL of Ec CFS, 50 μL of LTE, and 100 μL of HeLa reactions were compared. Comparing pull-down results for post-translationally captured proteins, we found the percentage of functional protein fractions to decrease from ~90% and ~70% in HeLa and LTE, respectively, to 50–60% in Ec CFS (Fig. 8c). In Ec CFS, the resin capture rescued a larger protein fraction from forming aggregates where around a half of captured protein material was present in misfolded or semi-aggregated form and required further on-resin denaturant and redox treatment to increase the proportion of functional protein to

80–90% (Fig. 8c, d). Finally, we used resin-assisted Ec CFS and LTE to produce CoV2-RBD, which could not be purified in soluble form from *E. coli* cells, and confirmed its interaction with anti-CoV2-RBD commercial antibodies (Fig. 8e). We further proved its native state in pull-down assay with ACE2 produced in LTE showing a twofold increase in the yield of a functional form following denaturant treatment (Fig. 8f).

## Discussion

Single-domain proteins above 100 aa in size fold under various degrees of kinetic control. The folding landscape of multidomain proteins is more complex, especially if it involves the disulfide-bond formation and the isomerization of prolyl bonds. With the loss of both intracellular compartmentalization and concerted chaperone activity in cell-free translation systems, accommodation of each unique folding landscape requires optimization of multiple system parameters. In this study, we describe the approach for the folding of in vitro produced (poly)peptides based on a corollary to "Anfinsen's dogma" and a physical theory of protein folding[56,58] suggesting that most evolutionary-selected (poly)peptide sequences in the aggregation-free environment can only adopt either fully folded or denatured conformations near thermodynamic equilibrium regardless of their folding landscape at physiological conditions (Fig. 1c). Although peptides being small enough are expected to fold under complete thermodynamic control even at physiological conditions[45], the folding of cysteine-rich peptides is kinetically dominated by covalent interactions that often result in disulfide-bond scrambling[34]. Therefore, instead of accommodating the folding constraints in each specific case by various additives, we take advantage of exquisite selectivity and affinity of biological interaction to co-translationally segregate the newly translated chains to affinity resin. By these means, we separate translation and protein folding to different reaction compartments partially avoiding a trade-off between folding and aggregation at the first stage (Fig. 1b). At the second stage, the immobilized (poly)peptides are transferred from the physiological environment, where the folding is partially kinetically controlled, to thermodynamically controlled folding space by adjusting the denaturant and/or reductant concentrations (Fig. 1c).

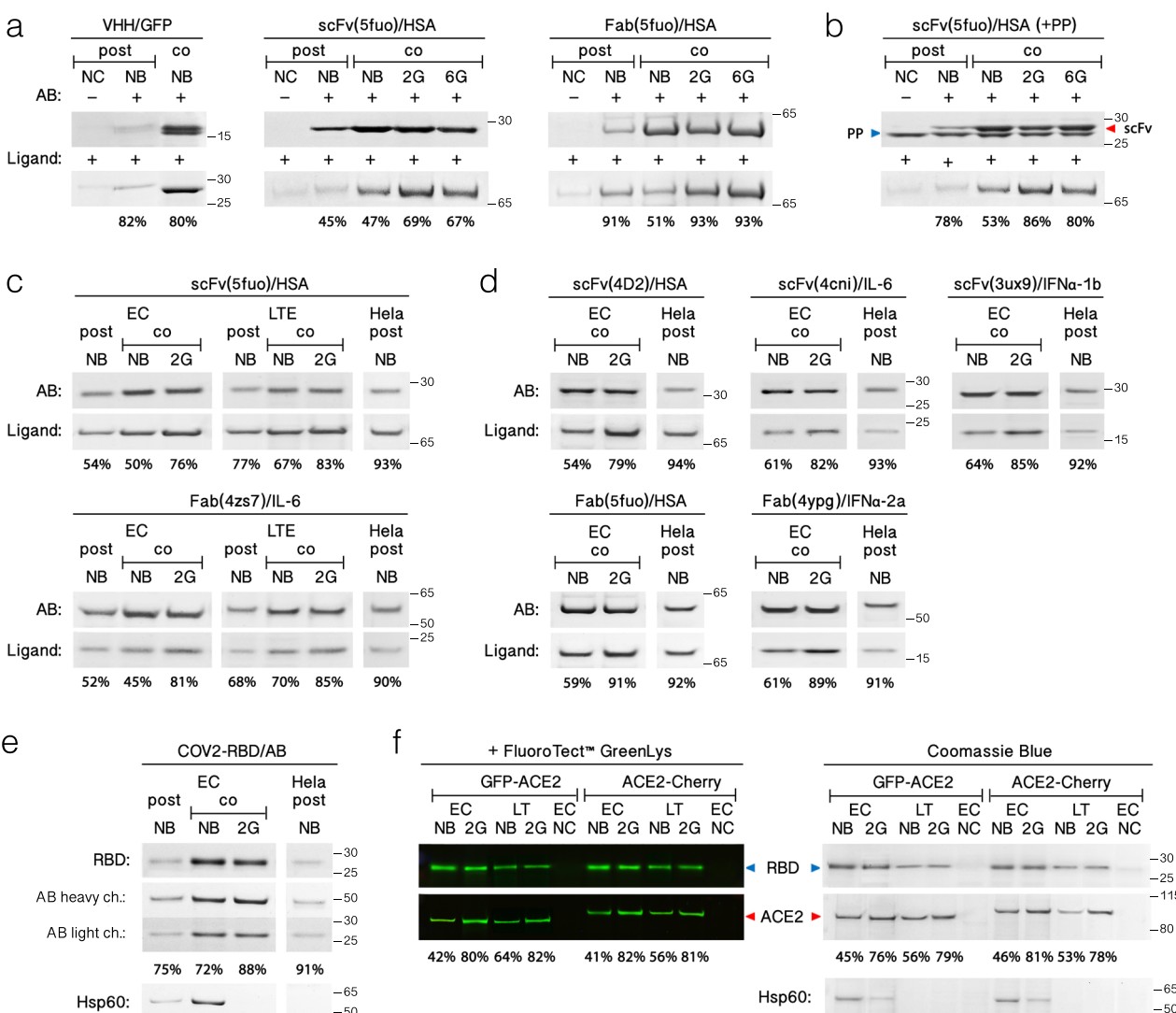

**Fig. 8 Analysis of folding efficiencies of disulfide-constrained proteins in different translation systems. a** Pull-down assay of different antibody fragments with respective antigens (both shown via slash on top of the gel stack) following co- (co) or post-translational (post) capture to Affinity-clamp-resin in Ec CFS. Translation reaction lacking template serves as negative control (NC). Prior to pull-down assay, resin-bound protein fractions were treated ON either with neutral buffer containing no denaturant (NB), or with the buffer system (2 G) containing 2 M of guanidine hydrochloride (Gdn) and 100 mM DTT for 2 h followed by 16 h incubation with 1 M Gdn and 10 mM GSH at RT. Alternatively, control samples underwent full off-resin refolding with 6 M GdnHCl (6 G). 'AB' and Ligand denote the presence antibody fragments translated and ligands used for pulldown assay, respectively. Percents of folded fractions were calculated following the equation (ligand density × $M_w^{AB}$)/($M_w^{Ligand}$ × AB density) and shown below each stack. The densities were obtained as non-saturated integration of the respective bands using the ImageJ software. **b** Same as in (**a**) but instead of heat-denaturing the AB/Ligand complexes were eluted by cleavage of antibody fragments with Prescission protease (PP) upstream to RGS-tag. **c**, **d** Same as in (**a**) but the yield and folding efficiency of antibody fragments obtained from resin-assisted Ec CFS are compared to antibody fragment production in conventional LTE and/or HeLa cell-based translation systems (ThermoFisher Scientific). **e** Same as in (**a**, **d**) but SarsCov2-RBD (COV2-RBD) was produced and probed with anti-SarsCov2-RBD antibody. **f** Same as in (**a**, **c**) but COV2-RBD was produced in resin-assisted Ec CFS supplemented with Lys-tRNA conjugated with BodiPy-FL (FluoroTect™ GreenLys, Promega) at 1:200 dilution and probed with ACE2- fusion to either GFP or mCherry following its synthesis in LTE supplemented with the same FluoroTect™ GreenLys dilution. The respective bands were visualized either by gel fluorescence scanning (left) following the elimination of both GFP and mCherry intrinsic fluorescence by boiling or by Coomassie-staining (right). Lower panels in e and f show bands corresponding to RBD protein-binder co-purified from Ec CFS and identified as Hsp60. Full gel scans are available in Supplementary Figs. 21 and 23 as well as in the Source Data file.

We show that supplementation of the translation reaction with affinity resin reduces the aggregation through the co-translational capture of non-productive folding states and/or high-energy intermediates. Disulfide-constrained peptides belonging to various structural classes were also shown to undergo various degrees of covalent aggregation in a standard resin-assisted reaction (Fig. 3d, e and Supplementary Fig. 7). Following co-translational resin capture, we attempted to recycle the non-

productive kinetically trapped folding states and/or folding intermediates under thermodynamic control. We demonstrated that treatment with 10 mM GSH for 12–48 h yielded productive oxidative folding on resin for all tested cysteine-rich peptides regardless of the number of disulfide bonds, embedded ring size or folding mode. Curiously, protein refolding on solid phase with 1.6–2 M GdnHCl was shown to be partially limited by cis/trans-peptidyl-prolyl isomerization when expressed in a resin-assisted

*E. coli*-based cell-free system (Ec CFS) but not in a resin-assisted *Leishmania* system (LTE). Since unfavorable trans-to-cis proline isomerization step can be catalyzed by PPIase only in cooperation with chaperones[60] the observed effect in LTE could be attributed to co-translational chaperon assistance inherent to eukaryotic translation system.

We have validated the developed approach by prototyping of disulfide-constrained proteins including scFv and Fab fragments at 50–200 µg/mL with the functional proportion amounting to 70–90% as estimated by pull-down assay (Fig. 8). Using the established platform, we also rapidly produced, purified, and assayed a number of therapeutically relevant peptides comprising cyclized or open backbones, ranging from a simple disulfide-bonded hairpin-like structures to complex knottins with intricate disulfide connectivity (Table 1). Taking into account, the universal applicability of the approach to a broad peptide range, peptide yield of 25–70 µg/ml from the batch reaction, and low cost of in-house prepared Ec CFS supplemented with the appropriate amount of affinity resin ($10/ml)[65], the total cost and timeline for the small-scale synthesis of the complex disulfide-constrained peptide can be reduced by ~tenfold compared to custom chemical synthesis from ~$2000/mg in 1–2 weeks to ~ $200/mg in 1–2 days, respectively.

Although the proposed strategy is likely to benefit from the integration with homeostatic energy regeneration[6] and the use of engineered strains[8,36] it is important to note that in its current format it is rather applicable for small-scale (poly)peptide prototyping due to limited reuse of the protein-coated resin. However, this approach can be potentially tailored to biomanufacturing via utilizing an inexpensive and recyclable metal-affinity resin instead. In contrast to protein-coated resin the latter would give an advantage of using elevated denaturant concentrations and prolonged incubation time to ensure complete resolution and recycling of all inter- and intramolecular aggregates. It may entail, however, the need for additional purification[42,54] and optimization of the reductant concentration to avoid metal reduction.

At last, we also adopted previously established in situ codon-reassignment techniques for simultaneous repurposing of two sense-codons and demonstrated the site-selective installation of two clickable amino acids followed by on-resin macrocyclization. The use of a crude Ec CFS, in alternative to expensive PURE system[63], and no protein carrier requirement in combination with emulsion-based microcompartmentation technique hold potential to convert the described approach to diversity-oriented screening platform, thereby allowing cost-effective selection of peptide binders against therapeutically relevant targets. Offering the standard procedure for prototyping of a variety of (poly) peptides the proposed strategy can aid in protein/peptide engineering by streamlining design–build–test cycles, a rapid iteration through which remains one of the major bottlenecks of synthetic biology.

## Methods

**Plasmid construction**. Sequence of the affinity clamp (PDZ-fibronectin fusion protein) was extracted from the PDB file 3CH8. Sequences of the anti-HSA scFv and Fab antibody fragments were derived from the PDB file 5FUO or from the US_2014_0186365 patent (sequence 4D2). Sequences of the anti-IL-6 scFv, anti-IL-6 Fab, anti-IFNα-1b scFv, and anti-IFNα-2a Fab were derived from the following PDB files: PDB:4CNI, PDB:4ZS7, PDB:3UX9, PDB:4YPG, respectively.

With minor exceptions, all coding sequences for peptides and proteins (Supplementary Table 3) were supplied as gBlock synthetic fragments by IDT (Integrated DNA Technologies), carrying ~30 bp vector-complementary flanks and subcloned via the Gibson Assembly method to the *Nco*I/*Not*I-opened pLTE (AddGene: 67044) or pOPINE (GenBank: EF372397.1) plasmids (Supplementary Table 3) according to the manufacturer's instructions (New England Biolabs). To create SFTI ORF with unique AGG and AGT codons to be reassigned to the encoding of unnatural amino acids, two cysteine codons in wtSFTI ORF were replaced with the respective AGG and AGT triplets using Q5® Site-Directed

Mutagenesis Kit (NEB) in two successive PCR rounds with oligonucleotides shown in Supplementary Table 6.

**Peptide quantification by AC-assay**. The peptide quantification assay based on activation of allosterically regulated TVMV protease in the context of peptide biosensor was established earlier in our lab[39]. Briefly, a clamping of C-terminal RGSIDTWV between ePDZ and Fn3 domains of affinity clamp results in dislodgment of the inhibitory peptide from the active site of TVMV, leading to protease activation and subsequent cleavage of the otherwise quenched reporter substrate. Typically, the affinity-clamp assay was carried out in a buffer containing 50 mM Tris-HCl, 1 M NaCl, 1 mM DTT, and 0.5 mM EDTA (pH 8.0), supplemented with 1 µM of peptide biosensor and 15 µM of TVMV-substrate peptide. The reaction progress was monitored by exciting the sample at 330 nm and recording the fluorescence changes at 405 nm for 3 h using the BioTek Synergy 4 Multi-Mode Microplate reader.

**Trypsin-inhibitory assay**. Trypsin was freshly prepared at 400 pM in the assay buffer containing 0.1 M NaCl, 10 mM CaCl₂, 0.005% Triton X-100, 0.1 M Tris-HCl pH 8.0. Trypsin at 100 pM was pre-incubated with peptide (twofold dilution series from 4 nM to 3.9 pM) in 200 µL of the assay buffer for 3 h at room temperature. After incubation residual trypsin activity was measured by adding the quenched fluorescent peptide substrate Boc-Q-A-R- 7-amido-4-methyl-coumarin to a final concentration of 10 µM. The fluorescence increase upon the release of 7-amido-4-methyl-coumarin was monitored on Infinite M1000 Pro plate reader (Tecan) every 30 s using excitation and emission wavelengths of 360 and 460 nm, respectively. To calculate IC$_{50}$, the initial velocities of the substrate hydrolysis by trypsin in the presence of different concentrations of peptide were fitted to a nonlinear regression curve using the software package Prism (GraphPad Software) followed by determining the corresponding $K_i$ value using a tight-binding equation[66] and a $K_m$ value of 12 µM.

**RGS-tag removal with trypsin and asparagine endopeptidase (AEP)**. Agarose-immobilized trypsin (treated with L-1-tosylamide-2-phenylethyl chloromethyl ketone (TPCK) to reduce the chymotrypsin activity) was purchased from ThermoFisher (#20230) and equilibrated with digestion buffer (0.1 M NH₄HCO₃, pH 8.0), aliquoted and stored at 4 °C as 50% slurry (vol/vol). 20 µL of mixed trypsin–resin suspension was transferred into the filter-bottom tube, spun at 5000 × g for 2 min and then mixed with SFTI-RGS peptide in digestion buffer containing 0.1 M NH4HCO3 buffer, pH 8.0 at 1:1 or 1:10 protease to peptide ratios followed by incubation for 3 h at 37 °C using the ThermoMixer (Eppendorf). The cleavage reaction was stopped by the addition of trifluoroacetic acid to 2.5% and vigorous shaking. Following quantification of prmSFTI by LC/MS using the calibration curve, the cleavage reaction was lyophilized and the peptide was resuspended in the appropriate volume of the assay buffer. For asparagine endopeptidase-mediated cleavage/cyclization of the reduced peptide, substrates containing the AEP-recognition motif were incubated at ≤280 µM with 12 µg/mL of OaAEP1b in the digestion buffer (50 mM CH₃COONa, 50 mM NaCl, 1 mM EDTA, pH 5.0) overnight at room temperature.

**RGS-tag removal with thrombin, carboxypeptidase A and Y, PreScission protease and hydroxylamine**. Thrombin cleavage was performed using 20 pmol of RGS-fusion peptide and 2 NIH units of thrombin (Sigma-Aldrich #T7009) in a thrombin cleavage buffer containing 50 mM Tris-HCl, 10 mM CaCl₂, pH 8.4 at 37 °C for 6–12 h.

Carboxypeptidase A was used in the resin-immobilized form (Sigma-Aldrich #C1261). 40 µL of 50% (~0.2 U) suspension was transferred to the filter-bottom tube and equilibrated with the cleavage buffer. Carboxypeptidase A cleavage was performed on 20 pmol of RGS-fusion peptide in a cleavage buffer containing 25 mM Tris-HCl, 500 mM NaCl, pH 7.5 at 37 °C for 6–16 h.

Yeast carboxypeptidase Y powder (Sigma-Aldrich, #C3888) was adjusted to 0.2 mg/mL with the cleavage buffer containing 50 mM Na₃C₆H₅O₇, pH 6.0. The cleavage reaction containing 2 pmol/µL of RGS-fusion peptide, 40 µg/mL of enzyme in the cleavage buffer was incubated at 37 °C for 0.5 h or overnight.

PreScission protease was expressed in *E. coli* and purified to homogeneity; the protease was adjusted to 4 mg/mL with storage buffer (50 mM Tris-HCl, 150 mM NaCl, 10 mM EDTA, 1 mM DTT, pH 8.0, 20% glycerol). The cleavage of resin immobilized protein/ligand complexes derived from 100 µL of resin-assisted translation reaction (~20 µL of settled AC-coated resin gel) was performed in 20 µL of buffer (50 mM Tris-HCl, 150 mM NaCl, pH 7.5) containing 20 µg/mL of PreScission protease at 4 °C for 16 h.

All cleavage reactions were terminated by the addition of an equal volume of 2.5% TFA. Sample aliquots withdrawn at different time points were purified using Pierce C18 Spin Tips (ThermoFisher) and concentrated by SpeedVac vacuum concentrator before analyzing by MALDI or LC-MS.

For hydroxylamine cleavage, AC-resin with 1–50 µg of bound peptide-RGS harboring accessible NG-cleavage motif was incubated at 45 °C for 4 h in cleavage buffer containing 2 M hydroxylamine in 15 mM Tris buffered to pH 9.3 with lithium hydroxide. After incubation, the flow-through containing cleaved peptide was collected by centrifugation at 3000×g and the beads were washed in six

alternating washing steps with washing buffer (50 mM Tris-HCl, 0.5 M NaCl, 0.1% Triton X-100, pH 7.5) and water followed by three final washings with water. The remaining resin-bound peptide was eluted with 0.2% TFA.

**Insecticidal activity assay.** The insecticidal activity of the Dc1a and its cell-free produced analog was tested by microinjection in female *Drosophila melanogaster* fruit flies aged 3–4 days with an average weight of 0.8–0.9 mg according to the previously described method[67]. Briefly, the C-terminally unprocessed Dc1a (Dc1a-GTGSGG-RGSIDTWV) was dissolved in water for injection, and water was used as the respective negative control. The thrombin-cleaved Dc1a sample (Dc1a-GTGSGG-R) contained thrombin and the respective amount of thrombin was added to the control sample. The injection volume was 50 nL and three repeats of seven doses (each at n = 8 fruit flies) were used for each Dc1a analog. Female *Drosophila* were cooled on ice before injection. At 24 h after the injection, the fruit flies were monitored for lethal effects and the median lethal dose (LD50) was determined. The number of control flies affected at the respective post-injection time was subtracted from the number of affected flies. The corrected numbers of affected *Drosophila* flies were upscaled to 100% and the resulting percentages were used to interpolate the respective LD50 values by fitting log dose–response curves using nonlinear regression analysis.

**Selective inactivation of specific endogenous tRNAs in *E. coli* S30 extract.** For inactivation of selected tRNAs, the *E. coli* S30 cell lysate was incubated with M5-1[25] and R7 antisense oligonucleotides (Supplementary Table 4) at 37 °C for 5 min to sequester endogenous tRNASerGCU and tRNAArgCCU, respectively. Both oligonucleotides were added to 9 μM final concentration from 400 μM stock solutions. Following incubations, the lysate was chilled on ice, supplemented with protease inhibitor cocktail (cOmplete™ Protease Inhibitor Cocktail, Roche) from 50× stock to achieve 1.7× final concentration and left on ice for 15 min prior to in vitro translation.

**Synthesis and purification of tRNAs.** DNA templates containing T7-promoter- and tRNA-coding sequences were assembled in 3-step PCR reactions as described[38] with minor modifications. Briefly, in step 1 forward oligonucleotide (F) containing T7-promoter sequence was combined with reverse oligonucleotide (R1) for overlap extension following two cycles of 1 min denaturation, 1 min 42 °C annealing and 30 s elongation. In step 2, 5′-GCGG-extended 3′-G-ending T7-promoter oligonucleotide was combined with R2-oligonucleotide spanning 3′-part of tRNA sequence for five amplification cycles with 1 min denaturing, 30 s 42 °C annealing and 15 s amplification. In step 3, both T7-promoter and R3-oligonucleotide complementary to 3′-part of tRNA were used at 7.5 μM concentrations for 28 amplification cycles with 30 s denaturing, 30 s annealing at 42 °C and 20 s elongation. After completion, PCR reactions were diluted 2.5-fold with water and the resulting PCR-products were purified by ethanol precipitation and dissolved in water. Run-off transcription using T7 RNA polymerase was performed for 3 h or overnight at 35 °C in a buffer containing 40 mM Hepes-KOH (pH 7.9), 20 mM Mg(OAc)$_2$, 2 mM Spermidine, 40 mM DTT, 5 mM of each rNTP, 0.25 μM DNA template, 10 μg/mL T7 polymerase, and 0.25 U/mL yeast inorganic pyrophosphatase. The tRNA transcripts were purified by affinity chromatography using ethanolamine–Sepharose matrix as described previously[38]. Briefly, for 1 mL of transcription reaction, 0.2 mL of the settled matrix was used. Following 1 h incubation at 4 °C, the resin-bound tRNAs were extensively washed with a buffer, containing 200 mM NaOAc pH 5.2, 0.25 mM EDTA. tRNAs were eluted from the matrix into the buffer containing 2 M NaOAc pH 5.2, 10 mM MgCl$_2$, 0.25 mM EDTA. tRNAs were ethanol precipitated and the pellets were dissolved in tRNA buffer containing 1 mM MgCl$_2$ and 0.5 mM NaOAc (pH 5.0). The sequences of tRNAs and oligonucleotides are summarized in Supplementary Tables 5 and 7.

**Purification of orthogonal aminoacyl-tRNA synthetases.** The protein sequences of orthogonal tRNA synthetases are provided in Supplementary Table 6. Engineered tyrosyl-tRNA synthetase from *Methanocaldococcus jannaschii* (AzFRS.2.t1) and pyrrolysine-tRNA-synthase from *Methanosarcina barkeri* (chPylRS) were expressed in BL21(DE3) RIL or Rosetta cells, respectively, and purified by Ni$^{2+}$ affinity chromatography and gel filtration as described previously. Briefly, the protein expression was induced with 0.5 mM isopropyl-β-D-thiogalactopyranoside at OD$_{600}$ 0.8 with the following overnight incubation at 20 °C. The cell pellet was resuspended in a binding buffer containing 50 mM sodium phosphate, pH 8.0, 0.3 M NaCl, 0.1 mM ATP, 5 mM 2-mercaptoethanol, and 20 mM Imidazole. Cells were disrupted using a continuous flow mode cell disruptor (Constant Systems) and proteins were purified on Ni$^{2+}$ affinity chromatography. Following affinity-chromatography, AzFRS.2.t1 was further purified by gel filtration on a Superdex 200 column (GE Healthcare) in PBS. In the case of chPylRSAF N-terminal 6 his-tag was cleaved off by PreScission protease. The gel filtration was performed on Superdex 200 equilibrated with 40 mM HEPES-NaOH pH 7.9, 100 mM KCl, 10 mM MgCl$_2$, 0.1 mM ATP, 1 mM TCEP, 10% (vol/vol) glycerol. Purified proteins were concentrated, snap-frozen in liquid nitrogen and stored at −80 °C in aliquots.

**In vitro production of the macrocyclic peptide with unnatural bond.** p-Azido-L-phenylalanine (AzF) and n-propargyl-L-lysine (PrK) were purchased from

SynChem and Sirius Fine Chemicals, respectively. AC-resin assisted in vitro translation reaction was assembled with S30 lysate treated with the antisense oligonucleotides and additionally supplemented with 12 μM tRNAAzF4-CCU, 10 μM AzFRS.2.t1, 1.5 mM AzF, 15 μM tRNAPylO2-ACU, 30 μM chPylRSAF, 1.5 mM Prk. The reaction was initiated by the addition of SFTI(AGG, AGT)-RGS coding template (Supplementary Table 3). The reaction was incubated at 32 °C for 90 min with shaking at 1400 rpm. The resin was washed as described above, and the peptide was either subjected to cyclization via copper-catalyzed azide-alkyne cycloaddition (CuAAC) on-resin or off-resin following elution with 0.2% TFA. For on-resin CuAAC, the reaction volume was adjusted with water to the initial translation reaction volume and further mixed with 1/7 volume of 1 M potassium phosphate (pH 7.4) and the same volume of catalytic premix consisting of 1 mM CuSO$_4$ and 5 mM of Cu$^{+1}$-stabilizing reagent BTTP. The reaction mixture was gassed with nitrogen for at least 1 min and the cyclization was initiated by adding 1/14 volume of freshly prepared 100 mM sodium ascorbate solution (Sigma-Aldrich) followed by 1 h incubation at room temperature. For analytical LC-MS analysis, the resin was washed and the cyclized peptide was eluted with 0.2% TFA. The sample was cleaned up using Zip-tip (Pierce C18 Spin Tips) according to the manufacturer's protocol and further eluted into 0.1% TFA, 80% ACN.

**Reverse-transcriptase-coupled quantitative real-time PCR (RT-qPCR).** RT-qPCR using a previously selected pair of oligonucleotides targeting 16S rRNA was employed for comparative quantification of ribosomal content in bacterial S30 extract-based and reconstituted (PURE) translation systems. To avoid genomic DNA interference at the PCR stage, the respective translation reactions were treated with DNase I in 20 μL volume containing 4 U of DNase I (NEB, #M0303S), 4 μL of 50-fold water-diluted translation mixture, and 2 μL of 10× DNase I buffer for 30 min at 37 °C. DNase I was inactivated by heating at 75 °C for 10 min, following the addition of EDTA to 2.5 mM final concentration. The resulting reaction was diluted fivefold with water and 2 μL was combined with an equal volume of 1 μM reverse oligonucleotide targeting rRNA. The annealing reaction was performed at 78 °C for 8 min and allowed to slowly cool to RT. For cDNA synthesis 4 μL of annealing reaction was supplemented with 1 μL of 10× buffer for Avian Myeloblastosis Virus reverse transcriptase (AMV RT), 0.25 μL of dNTP mix (10 mM each), and 2 U of AMV RT (NEB, #M0277). The cDNA synthesis was performed at 50 °C for 45 min followed by 85 °C for 5 min for inactivation of reverse transcriptase. For qPCR, 1.25 μL or 2.5 μL of RT reaction was added into the well of 384-well plate (PerkinElmer, #6007290) containing 11.2 μL or 10 μL, respectively, of premix composed of 6.25 μL of Platinum™ SYBR™ Green qPCR SuperMix-UDG (Thermofisher), 0.25 μL of each oligonucleotide (10 μM stock concentration), 0.025 μL of ROX dye (ThermoFisher), 0.075 μL of DMSO, and 5 or 3.75 μL of water, respectively. The standard cycling program (95 °C, 12 min for initial denaturing, 40 cycles of 95 °C for 15 s, 60 °C for 1 min) was used followed by melting curve analysis using the default program of Applied Biosystems® ViiA™ 7 Real-Time PCR System.

**Preparation of affinity-clamp-coupled resin (AC-resin).** The affinity-clamp protein (PDZ-fibronectin fusion protein, PDB: 3CH8) harboring C-terminal cysteine residue (Supplementary Table 6) was purified from BL21(DE3)RIL by Ni$^{2+}$-affinity chromatography followed by gel filtration. Briefly, following induction with 0.5 mM isopropyl-β-D-thiogalactopyranoside (IPTG) at OD$_{600}$ 0.8, protein expression was carried out ON at 20 °C. Cells were disrupted using a continuous flow mode cell disruptor (Constant Systems) and protein was purified by Ni$^{2+}$-affinity chromatography using standard buffers followed by gel-filtration chromatography on a Superdex 75 column (GE Healthcare) in 50 mM Tris-HCl (pH 8.0), 0.1 M NaCl, 1 mM EDTA, 2 mM TCEP. The protein sample was adjusted to 10 mg/mL in coupling buffer (50 mM Tris-HCl, 5 mM EDTA, 25 mM TCEP-KOH (pH 8.0), pH 8.5) and added to saturation to UltraLink® iodoacetyl resin (ThermoFisher, #53155) pre-washed with the coupling buffer. Following 1 h coupling reaction at room temperature, the resin was settled, washed with coupling buffer, blocked with 50 mM L-cysteine for 30 min. The AC-coated resin was washed again with 1 M NaCl and PBS and stored in PBS containing 0.1 mg/ml of BSA and 2 mM NaN3 as 50% (vol/vol) suspension at 4 °C. By titration of the resin amount derived from 50 μL of 50% (vol/vol) suspension with RGSIDTWV peptide, the binding capacity of the AC-resin was found to be 27 nanomoles of ligand per milliliter of settled resin gel or per 2 mL of 50% (vol/vol) suspension (Supplementary Fig. 1). For details, refer to a procedure overview in the Supplementary Methods.

**Preparation of *E. coli* S30 extract and translation reaction assembly.** For details, refer to a procedure overview in the Supplementary Methods. Briefly, *E. coli* S30 extract was prepared as described by Schwarz et al.[68] with minor modifications. Briefly, cells were cultivated in 5–10 L of filter-sterilized TBGG media (tryptone 12 g/L, yeast extract 24 g/L, glycerol 8 mL/L, glucose 1 g/L, KH$_2$PO$_4$ 2.31 g/L, K$_2$HPO$_4$ 2.54 g/L) to OD 3.5, pre-chilled with 5 × 200 or 10 × 200 mL of −80 °C pre-frozen packs of LB broth and spun at 2.5 kg for 15 min. The cell pellet from the final wash was resuspended in 200 % (vol/wt) of S30B buffer and disrupted using fluidic disruption (Constant Systems, continuous flow mode) at 20 kpsi at 4 °C. Cell homogenate was cleared by two consecutive 30 min centrifugation steps at 30 kg at

4 °C while collecting the top ¾ of supernatant every time. The final supernatant was adjusted to 0.4 M with 5 M NaCl followed by 45 min incubation at 42 °C in a water bath. Following incubation, the cell homogenate was transferred to dialysis tubing (12–14 kDa cutoff, SpectrumTM Labs) and dialyzed for 2 h against 4 L of cold S30C buffer at 4 °C followed by ON dialysis against the same volume of fresh buffer. The dialyzed extract was centrifuged for 30 min at 30 kg at 4 °C and top ¾ of supernatant was collected, aliquoted, and frozen in LN2 for −80 °C storage. The translation reaction was assembled using 35% (vol/vol) S30 extract (in 10 mM Tris-acetate pH 8.2, 14 mM Mg(OAc)$_2$, 0.6 mM KOAc, 0.5 mM DTT, 40% (vol/vol) of ×2.5 Feeding Solution (236 mM HEPES-KOH pH 7.4, 12.5 mM Mg(OAc)$_2$, 375 mM KOAc, 5% PEG 8000, 12.5% glycerol, 5 mM NaN$_3$, 0.015% Tween 20, 5 mM DTT, 2.5× protease inhibitor (cOmplete™ EDTA-free, Roche). 0.25 mg/mL folinic acid, 2 mM of each rNTP with an extra 1 mM of ATP, 38 mM of acetyl phosphate, 68 mM of creatine phosphate, 1.25 mM of each amino acid with extra 2.5 mM for Arg, Cys, Trp, Asp, Met, Glu), 0.05 mg/mL T7 RNA polymerase, 45 U/mL creatine phosphokinase, 0.05 mg/mL of TVMV protease and 20 nM of plasmid template. Translation of disulfide-rich proteins was performed in reactions supplemented with the extra 7.5 mM DTT and constituted with 17.5% (vol/vol) of S30 extract and 17.5% (vol/vol) of the extract buffer. Cyclosporin A (Sigma-Aldrich, #PHR1092) or Rapamycin (Sigma-Aldrich, #A3782) were supplemented to the respective translation reactions at 10 μM final concentration.

**Preparation of *Leishmania* cell-extract and translation reaction assembly**. A *Leishmania*-based transcription–translation system (LTE) was prepared as described by Kovtun et al.[40]. Briefly, *Leishamnia tarentolae* cells were expanded in 5-L conical flasks at 26.5 °C, 74 rpm agitation in a 1 L per flask of TBGG media (as for S30 extract, supplemented with hemin). Cells were harvested at 1.0–1.2 × 10$^8$ cells/ml, pelleted at 2.5 kg, and resuspended to 10$^{10}$ cells/ml in a buffer containing 45 mM HEPES-KOH pH 7.6, 250 mM Sucrose,100 mM KOAc, 3 mM Mg(OAc)$_2$ followed by disruption in a nitrogen cavitation device (70 bar N$_2$, 45 min equilibration at 4 °C). Following two sequential 10,000 × g and 30,000 × g centrifugation, the top 2/3 of the final supernatant was collected and subjected to gel filtration on PD-10 Superdex 25 column (GE Healthcare) into fresh elution buffer (EB; 45 mM HEPES-KOH pH 7.6, 100 mM KOAc, 3 mM Mg(OAc)$_2$). The 2.5 V of buffer-exchanged lysate was then supplemented with 1 V of 5x feeding solution containing 6 mM ATP, 0.68 mM GTP, 22.5 mM Mg(OAc)$_2$, 1.25 mM spermidine, 10 mM DTT, 200 mM creatine phosphate, 100 mM HEPES-KOH pH 7.6, 5% (vol/vol) PEG 3000, 5.25× protease inhibitor cocktail (Complete™ EDTA-free, Roche), 0.68 mM of each amino acid, 2.5 mM rNTP mix (ATP, GTP, UTP and CTP), 0.05 mM anti-splice leader DNA oligonucleotide (αSL oligo, Supplementary Table 7), 0.5 mg/ml T7 RNA polymerase, 200 U/ml creatine phosphokinase, snap-frozen and stored at −80 °C. Transcription–translation reaction was assembled by adjusting 30 μl of supplemented lysate to 100 μl final reaction volume with addition of plasmid template to final 20–40 nM, extra 5% of glycerol, 2 mM NaN$_3$, and 0.005% Tween 20.

**Translation in PURE and HeLa-based cell-free systems**. Protein synthesis in a fully reconstituted PURE (Protein synthesis Using Recombinant Elements) translation system was performed using PURExpress In Vitro Protein Synthesis Kit (NEB, #E6800S) according to the manufacturer's protocol. HeLa-based translation system was assembled using 1-Step Human Coupled IVT Kit (ThermoFisher, #88881) according to the manufacturer's instructions. HeLa-based translation reactions were performed in 100 μL at 30 °C for 6 h.

**Assembly of resin-assisted translation reactions**. The resin-assisted translation was performed for Ec CFS, PURE, and LTE. For details regarding the assembly of AC-resin-assisted Ec CFS also refers to a procedure overview in Supplementary Methods. Briefly, the amount of AC-coated resin corresponding to 40 μl of 50% (vol/vol) resin suspension in PBS was used to supplement 100 μl of transcription–translation reaction either co- or post-translationally. Prior to reaction in PURE and Ec CFS, the resin was washed and equilibrated with the reaction buffer consisting of 35% (vol/vol) of the S30-extract buffer and 40% of the feeding solution. For LTE reaction the resin was equilibrated in a buffer consisting of 21% EB and 8.5% of LTE feeding solution. For post-translational product pulldown, AC-coated resin was supplemented in one reaction volume of a buffer containing 20 mM Tris-HCl pH 7.5, 1 M NaCl, 0.05% Tween 20 to terminate the translation reaction and allow product capture. In parallel, resin-assisted translation reaction was supplemented with 1×V of the same buffer and both mixtures were incubated for another 1 h at RT with agitation. Following incubation, the mixtures were transferred to filter-bottom tubes, flow-throughs were separated and the resins carrying immobilized peptide translation products were washed with six alternating sessions of water and Wash Buffer (50 mM Tris-HCl, 500 mM NaCl, 0.1% Tween 20, pH 7.5) while for immobilized protein products the water was replaced with Neutral Buffer (NB) (20 mM Tris-HCl, 20 mM NaCl, pH 7.5).

Strep-Tactin-resin-assisted translation reactions were prepared in the same way as described for AC-coated resin, with minor exceptions. Briefly, the amount corresponding to 20 μL of 50% (vol/vol) suspension of Commercial Strep-Tactin XT (IBA Lifesciences) resin was used to supplement 100 μL of translation reaction (ligand-binding capacity of Strep-Tactin-coated resin was found to be 50 nmol of

ligand per ml of settled resin gel, corresponding to 2 ml of 50% (vol/vol) resin suspension in PBS). For post-translational product pulldown, following the completion of translation reaction, Strep-Tactin-coated resin was supplemented into translation in one reaction volume of a buffer containing 20 mM Tris-HCl pH 7.5, 100 mM NaCl, 0.05% Tween 20. In parallel, resin-assisted translation reaction was supplemented with 1xV of the same buffer and both mixtures were incubated for another 1 h at RT with agitation. The same buffer was used as a Wash Buffer in combination with NB (20 mM Tris-HCl, 20 mM NaCl, pH 7.5) for six alternating washing sessions. Captured products were eluted from Strep-Tactin-coated resin with elution buffer (100 mM Tris-HCl pH 8.0, 150 mM NaCl, 0.25 mM EDTA) containing 50 mM biotin in two consecutive steps with 15 min incubation at 30 °C with 1400 rpm agitation at each step.

**Peptide translation in resin-assisted CFSs**. Peptide synthesis was performed at 27 °C for 3 h in LTE, and at 32 °C for 3 h in Ec CFS and 6 h in the PURE translation system. Affinity-clamp-assisted translation reactions were assembled and processed as described in the previous "Methods" section (for details also refer to a procedure overview in the Supplementary methods). RGS-tagged peptides were eluted from the AC-resin either with DMSO or 0.2% TFA, or a mixture of both at indicated proportions. The resin was incubated with 1 V of elution buffer relative to 50% suspension for 20 min with vigorous shaking at RT followed by centrifugation for 2 min at 17 kg. The elution step could be repeated three times followed by snap freezing and lyophilization of combined elution. To perform on-resin peptide reduction the resin was incubated with 50 mM DTT in 0.1 M NH$_4$HCO$_3$, pH 8.5 at RT for 30 min followed by washing and oxidative folding in a buffer containing 10 mM or less glutathione (GSH) in 0.1 M NH$_4$HCO$_3$ pH 8.5 at RT for 12 or 48 h with exception for kalata B1 (Supplementary note 6). For further analysis, peptides were eluted with 1 V of 0.2% TFA in three elution sessions followed by lyophilization. For more details, refer to a procedure overview in the Supplementary Methods.

**Translation and on-resin refolding for proteins free of disulfide bonds**. Translation was performed in Ec CFS and LTE in 100 μl reaction volume at 28 °C and 27 °C for 3 h, respectively. Affinity-clamp- and Strep-Tactin-assisted translation reactions were assembled and processed as described above under the "Assembly of resin-assisted translation reaction". Proteins were eluted from Strep-Tactin-resin with an elution buffer (100 mM Tris-HCl pH 8.0, 150 mM NaCl, 0.25 mM EDTA) containing 50 mM biotin. Resin amount corresponding to 6 μL of 50% suspension (vol/vol) was eluted with 20 μL of buffer in two consecutive elution steps. For refolding of Strep-Tactin-resin-immobilized proteins the given amount of resin was incubated with 40 μl of refolding buffer (20 mM Tris-HCl, 2.5 mM DTT, 20 mM NaCl, 0,125% Tween 20, pH 7.5) containing either 0.8 or 1.6 M of guanidine hydrochloride (GndHCl) for 2 h at RT with 1400 rpm agitation in top-bench thermomixer (Eppendorf). Guanidine hydrochloride was originally prepared as 8 M stock based on neutral buffer. Treatment of Affinity-clamp-resin-immobilized proteins was performed in a buffer (20 mM Tris-HCl, 2.5 mM DTT, 1 M NaCl, 0,125% Tween 20, pH 7.5) containing 2 M GdnHCl for 6 h or 12 h at RT with 1400 rpm agitation. Control resins were incubated in the neutral buffer (NB) containing 20 mM Tris-HCl, 2.5 mM DTT, 20 mM NaCl, 0,125% Tween 20, pH 7.5. Following incubation, the refolding reactions were diluted twofold in two consecutive dilution steps with 1 h incubation following each dilution. Finally, the resins were extensively washed with the neutral buffer (20 mM Tris-HCl, 20 mM NaCl, 0.05% Tween 20, pH 7.5). For details, refer to a procedure overview in the Supplementary Methods.

**AC-assisted translation, on-resin refolding, and pull-down analysis for disulfide-rich proteins**. The antibody fragments and the CoV2-RBD (Supplementary Table 3) were translated in 20 μl of Ec CFS, 50 μl of LTE at 25 °C for 4 h and in 100 μl of HeLa-based translation system at 30 °C for 6 h. Ec CFS was modified with adjustment of DTT concentration to 10 mM and the use of S30 extract at twofold dilution. For details also refer to a procedure overview in the Supplementary Methods. Ec CFS and LTE translation reactions (performed in 20 μl and 50 μl, respectively) were supplemented with AC-coated resin amount corresponding to 8 μL and 20 μL of 50% (vol/vol) resin suspension co- and post-translationally while only post-translational pulldown was performed following translation in HeLa-based translation system. On-resin refolding for proteins immobilized co-translationally was performed by incubation of resin-bound protein fractions with a buffer containing 2 M guanidine hydrochloride (GdnHCl), 50 mM Tris-HCl, 1 M NaCl, 100 mM DTT, pH 7.5 for 2 h at RT followed by exchange into the buffer containing 1 M GdnHCl, 50 mM Tris-HCl, 0.5 M NaCl, and 10 mM reduced glutathione (GSH), pH 7.5, followed by 16 h incubation at RT with 1400 rpm agitation. Guanidine hydrochloride was originally prepared as 8 M stock based on neutral buffer. Control protein samples were incubated with a neutral buffer (NB) containing 20 mM Tris-HCl, 20 mM NaCl, pH 7.5. Following 16 h incubation, the mixtures were diluted by NB in two consequent twofold dilution steps and additionally incubated at RT for 1 h after each dilution step followed by the final resin washing step. Full denaturing/renaturing of antibody fragments was carried out in a buffer containing 6 M GdnHCl, 50 mM TrisHCl, pH 7.5, 1 M NaCl, and 100 mM DTT for 2 h at room temperature followed by

overnight dialyzing against buffer lacking denaturant (50 mM Tris-HCl, 0.5 M NaCl, pH 7.5) at 4 °C. For details refer to a procedure overview in the Supplementary Methods. Pull-down assay was performed with 50 µL of 10 µM solution of respective antigens such as GFP (purified from pOPINE in-house), Human serum albumin (Sigma-Aldrich, #A3782), human IFNα-1b (GenScript, #Z02866), human IFNα-2a (Shenandoah Biotechnology, #100-54-100UG), human IL-6 (Shenandoah Biotechnology, #100-10-100UG) or SARS-COV2-RBD antibody (Sanyou Biopharmaceuticals, #AHA004). Incubation was conducted at RT for 3 h. After washing with neutral buffer, the beads were boiled at 95 °C for 5 min with 2×LDS sample buffer (Invitrogen™ 4X Bolt™, #B0008) and eluted fractions were analyzed on SDS-PAGE.

**DHFR activity assay**. Fluorescence DHFR activity assay was performed by monitoring the change in NADPH fluorescence (Ex 340 nm, Em 375 nm) upon its conversion to $NADP^+$ in 200 µL of buffer containing 20 mM Tris-HCl pH 7.5, 100 mM NaCl, 1 mM Mg(OAc)$_2$, 0.012% Tween 20, 2.5 mM DTT in a black 96-well plate (Corning). Reactions were started simultaneously by pipetting the substrate mixture containing NADPH and dihydrofolic acid to 80 µM and 70 µM final concentration, respectively. Fluorescence changes were recorded for 4 h. Protein fractions bound to affinity-clamp-coated resin amount corresponding to 12.5 µL of 50% suspension (vol/vol) were directly used in the assay. Protein fractions from Strep-Tactin-assisted translation reactions were eluted from the resin corresponding to 6 µL of 50% suspension in two consecutive 20 µL elution steps (40 µL total elution volume). Pure recombinant *E. coli* DHFR was used to calibrate the assay (Supplementary Fig. 16B). Initial rates were derived from the fitting of the respective kinetic curves with polynomial or linear functions in Excel.

Colorimetric DHFR activity assay was used to monitor the activity of DHFR-calmodulin biosensor in response to a peptide inhibitor (Supplementary Note 8). The assay was performed by monitoring the decrease in absorbance of NADPH at 340 nm at 25 °C in 1 mL of 20 mM NaCl, 20 mM KH$_2$PO$_4$ pH 7.5 buffer containing 10 nM of chimeric DHFR-calmodulin sensor, 0.2 µM of M13 calmodulin-binding peptide, 80 µM NADPH, 67 µM dihydrofolic acid, 100 µM CaCl$_2$ and 2 µM of either reduced or oxidized HT-1-RGS fusion. The changes in absorbance were recorded on Cary 50 UV–Vis spectrophotometer (Varian Inc.). Only oxidized in vitro translated HT-1 inhibited the biosensor confirming the installation of native disulfide bridges critical for HT-1 functionality (Supplementary Fig. 11C).

**Peptide mass spectrometry**. LC-MS/MS analysis was performed on Nexera UHPLC (Shimadzu, Japan) interfaced with a TripleTOF 5600 mass spectrometer (ABSCIEX, Canada) equipped with a duo electrospray ion source. Lyophilized peptide 100–200 ng was dissolved in 20 µL of 0.1% formic acid. 15 µL of peptide sample was injected onto a 2.1 × 100 mm Zorbax C18 1.8 µm column (Agilent) for analytical chromatography. Chromatography was performed in the eluent system of 0.1% formic acid in water (A) and 0.1% formic acid in 90% acetonitrile (B), at 0.2 mL/min flow rate using the following pump settings: 50 min 1–40% B, 8 min 40–98% B, 3 min 98% B, 3 min 98-1% B. The following settings were used: the ion spray voltage was set to 5500 V, de-clustering potential - to 100 V, curtain gas flow - to 25, nebulizer gas 1 - to 50, nebulizer gas 2 - to 60, interface heater - to 150 °C, and the turbo heater - to 500 °C. The mass spectrometer acquired 200 ms full-scan TOF-MS data followed by up to 10200 ms full-scan product-ion data acquisition in an Information Dependent Acquisition mode. Full-scan TOF-MS data were acquired over the mass range 400–7000 and for product-ion MS/MS 200–3500. Ions observed in the TOF-MS scan exceeding a threshold of 100 counts and a charge state of +2 to +5 were set to trigger the acquisition of product-ion MS/MS spectra of the resultant 10 most intense ions. The data were acquired and processed using Analyst TF 1.6 software (ABSCIEX, Canada).

**MALDI-TOF**. Peptide samples were initially analyzed using the 5800 MALDI-TOF/TOF Mass Spectrometer (Applied Biosystems). Peptides were made up in 70% acetonitrile with 0.1% formic acid in water and mixed 1:1 with α-cyano-4-hydroxycinnamic acid matrix (10 mg/mL) in the same solvent. Then 0.8 µL (10–100 ng) of each sample was spotted onto a stainless-steel target and allowed to air dry. Spectra were obtained in positive ion, linear mode using an accelerator voltage of 20 kV, grid voltage of 64%, and delay time of 350 ns. Each spectrum consisted of 200–500 shots from random target positions. Spectra were processed using Data Explorer software.

**Western blot protocol**. Western blot was performed as described previously[69] with minor modifications. Briefly, protein samples were heated at 95 °C for 5 min with 2xSDS loading buffer (Invitrogen) and resolved on 12% SDS-PAGE (Invitrogen). Following electrophoresis, the proteins were transferred to PVDF membrane 0.2 µm (Millipore). Primary mouse anti-GFP antibodies (dilution WB 1:1000) (#11814460001, Roche) and secondary goat anti-mouse IgG (H + L) conjugated with HRP (dilution WB 1:5000) (Cat. No. G-21040, Life Technologies) were used for immunoblotting. Bound primary antibodies were visualized with horseradish peroxidase-conjugated secondary antibodies and the Super Signal West Dura ECL detection reagent (Life Technologies). The specific protein bands were visualized using the Super Signal West Dura ECL detection reagent (Life

Technologies) and imaged using the ChemiDoc Imaging System (Bio-rad, Gladesville, New South Wales, Australia).

**Reporting summary**. Further information on research design is available in the Nature Research Reporting Summary linked to this article.

## Data availability
The authors declare that the data supporting the findings of this study are available in this paper and its Supplementary Information files. The sequences of antibody fragments used in this study are available in the Protein Data Bank under accession codes 5FUO, 4ZS7, 4CNI, 3UX9, 4YPG. Source data are provided with this paper.

## Code availability
This study did not generate any new code.

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

## Acknowledgements

Authors are grateful to Dr. Kerry-Ann McMahon for her invaluable assistance with Western Blot performance as well as to Dr. Thomas Durek and Kuok Yap (Institute for Molecular Bioscience, University of Queensland) for discussion and provision of asparagine endopeptidase. The authors would like to acknowledge Dr. Patricia Walden for excellent organizational assistance and Dr. Wayne Johnston (Queensland University of Technology) for the provision of Leishmania-cell extract and accessory components for LTE translation reaction. This work was supported in part by the Australian Research Council Linkage Project LP150100689 and ARC Centre of Excellence in Synthetic Biology grant CE200100029. K.A. gratefully acknowledges the financial support of QUT-CSIRO Synthetic biology alliance. This work was also supported in part by contract research funding provided by Molecular Warehouse Inc. G.F.K. was funded by the NHMRC Principal Research Fellowship (APP1136889). V.H. was funded by an Australian Research Council Future Fellowship (FT190100482).

## Author contributions

Y.W. designed experiments, performed experiments, analyzed the data, and wrote the manuscript. Z.C. designed experiments, performed experiments, analyzed the data, and wrote the manuscript. Y.H. performed experiments, analyzed the data. S.J.V. performed experiments and analyzed the data. A.V.A. performed experiments. Z.G. designed experiments and performed experiments. S.V.M. performed experiments. A.O.H. performed experiments and analyzed the data. J.R.D. performed experiments and analyzed the data. S.G. performed experiments. K.E.C. performed experiments. B.M.C. designed experiments and analyzed the data. I.V. designed experiments and analyzed the data. V.H. designed experiments, analyzed the data, and wrote the manuscript. A.J. performed

experiments. M.A.C. designed experiments and analyzed the data. G.F.K. designed experiments, analyzed the data, and wrote the manuscript. D.J.C. designed experiments, analyzed the data, and wrote the manuscript. K.A. designed experiments, analyzed the data, and wrote the manuscript. S.M. designed experiments, performed experiments, analyzed the data, and wrote the manuscript. All authors discussed the results, refined the manuscript, and approved its final version.

## Competing interests

The authors declare no competing interests.
