## [Peer Review File · Nature Communications]

Reviewers' Comments:

Reviewer #1:

Remarks to the Author:

The authors have done a good job in revising the paper to address our earlier concerns. We recommend the revised paper for publication in *Nature Communications*.

Reviewer #2:

Remarks to the Author:

The authors should be commended for the thoroughness of their response. I appreciate the effort and support the revised manuscript.

Reviewer #3:

Remarks to the Author:

Cell-free peptide and protein synthesis, based on *in vitro* transcription and translation, is attracting a lot of attention right now. Many laboratories are using this technology because it enables a broad variety of bioengineering applications. There is, currently, a considerable effort put in this technology for biomanufacturing high-value peptides or proteins. In their article 'Towards a generic prototyping approach for therapeutically-relevant peptides and proteins in a cell-free translation system', Wu et al show that by using affinity matrix directly in cell-free reactions, one can produce greater amounts of bioactive peptides and proteins (such as antimicrobials and antibodies) compared to standard batch reactions. The authors show that with the affinity matrix, the level of aggregation of the synthesized peptides or proteins is reduced and that one can perform some folding biochemistry. The work covers a good set of peptides and proteins, including some related to the covid-19 virus. The experiments focus mostly on peptides and proteins with disulfide bonds. The study also includes the use of non-canonical amino acids. My comments are grouped into major and minor points.

Major points:

1. The use of an affinity matrix is relatively new and brings some originality to the current cell-free expression technology capabilities. The authors show that attaching a peptide or protein using an affinity tag onto a matrix simultaneously with its synthesis does improve the production of high-value biologics. In that sense, the work grasps the current trend very well, especially when it is about antimicrobial peptides and antibodies. Whether the set of proteins and peptides tested is enough to claim generalization of the approach is not clear, as in bioengineering, and any biology-related field, one can always ask for more tests. I personally think that the peptides and proteins assayed in this work form a reasonable set of tests. The authors support their observations with some good references. Especially, their analysis of the thermodynamics and kinetics of folding is interesting and they certainly provide some valuable insights into a complex problem.
2. The claim that this approach is simple is still not convincing. The work is very technical and requires substantial preparation. Claiming that it can be easily adopted by non-specialized labs is hard to believe. Cell-free expression systems are never easy to work with at the entry-level. It does not diminish the good observations made in the study. It just does not seem that easy for non-specialized labs. The preparation of the matrix, for instance, is facile when one knows how to do it, certainly less straightforward for non-specialists. Related to this, there are a lot of data, and it is hard to get clear messages from it. The number of abbreviations is very large, and it is often hard to follow, and some abbreviations are not spelled out. I suggest making a list of

abbreviations. Also, it would be good to summarize the method in a single figure or a protocol-like page. If it is that easy, it should be possible to summarize the approach concisely so that others can use it.

3. The results obtained in this work could have been discussed or compared with the current state-of-the-art capabilities of the field. For instance, some companies are selling cell-free systems specifically for producing peptides and proteins with disulfide bonds (e.g., SHuffle kit from the company NEB). Some of the results obtained with a lysate-based *E. coli* cell-free system are compared to the PURE system which is good. Another area that could have been included in the discussion is the current trend for fast biomanufacturing at the point of care. For example, several groups have shown how to prepare lyophilized cell-free proteins synthesis systems, removing the need for a cold chain and enabling biomanufacturing at the point of care. How does the approach described in this article position itself with respect to that? If the approach is not yet adapted for point-of-care applications, just state it, and explain how it could be adapted for point-of-care applications (if it is possible).

Minor points:

- Using cell-free translation all over the place is confusing, as it appears that the work is cell-free transcription and translation for the most part.
- Methods: should be reviewed and completed. For example, what is the composition of buffer S30B?

Reviewer #3 (Remarks to the Author):

Cell-free peptide and protein synthesis, based on in vitro transcription and translation, is attracting a lot of attention right now. Many laboratories are using this technology because it enables a broad variety of bioengineering applications. There is, currently, a considerable effort put in this technology for biomanufacturing high-value peptides or proteins. In their article 'Towards a generic prototyping approach for therapeutically-relevant peptides and proteins in a cell-free translation system', Wu et al show that by using affinity matrix directly in cell-free reactions, one can produce greater amounts of bioactive peptides and proteins (such as antimicrobials and antibodies) compared to standard batch reactions. The authors show that with the affinity matrix, the level of aggregation of the synthesized peptides or proteins is reduced and that one can perform some folding biochemistry. The work covers a good set of peptides and proteins, including some related to the

covid-19 virus. The experiments focus mostly on peptides and proteins with disulfide bonds. The study also includes the use of non-canonical amino acids. My comments are grouped into major and minor points.

Major points:

Point 1. The use of an affinity matrix is relatively new and brings some originality to the current cell-free expression technology capabilities. The authors show that attaching a peptide or protein using an affinity tag onto a matrix simultaneously with its synthesis does improve the production of high-value biologics. In that sense, the work grasps the current trend very well, especially when it is about antimicrobial peptides and antibodies. Whether the set of proteins and peptides tested is enough to claim generalization of the approach is not clear, as in bioengineering, and any biology-related field, one can always ask for more tests. I personally think that the peptides and proteins assayed in this work form a reasonable set of tests. The authors support their observations with some good references. Especially, their analysis of the thermodynamics and kinetics of folding is interesting and they certainly provide some valuable insights into a complex problem.

Response to point 1: We thank the Reviewer for the positive assessment of our work.

Point 2. The claim that this approach is simple is still not convincing. The work is very technical and requires substantial preparation. Claiming that it can be easily adopted by non-specialized labs is hard to believe. Cell-free expression systems are never easy to work with at the entry-level. It does not diminish the good observations made in the study. It just does not seem that easy for non-specialized labs. The preparation of the matrix, for instance, is facile when one knows how to do it, certainly less straightforward for non-specialists. Related to this, there are a lot of data, and it is hard to get clear messages from it. The number of abbreviations is very large, and it is often hard to follow, and some abbreviations are not spelled out. I suggest making a list of abbreviations. Also, it would be good to summarize the method in a single figure or a protocol-like page. If it is that easy, it should be possible to summarize the approach concisely so that others can use it.

Response to point 2: We agree with the Reviewer that for the research groups being new to cell-free translation the described approach would not seem trivial. However, *E. coli*-based cell-free system has an important advantage over the other cell-free systems especially of eukaryotic origin in that it is not very capricious owing to a less strict translational control. We also agree that the abundance of data obscures the technicalities of the method described. To

this end, following the reviewer's suggestion, we provide a step-by-step workflow describing the preparation of affinity clamp-coated resin, preparation of S30 extract, reaction assembly and refolding conditions for peptides, disulfide-constrained proteins and also for proteins free of disulfide bonds. The protocol page is now provided under the Supplementary methods section. We believe that ordering of the technical steps into a straight and cohesive procedure would make for a great complement to this work. We also included the list of abbreviations as Supplementary Table 8.

Below we provide the screenshot of the Supplementary methods section:

Supplementary methods

A. Preparation of affinity-clamp-coated resin

- The affinity clamp protein harboring C-terminal cysteine (Table 6) is adjusted to 10 mg/ml in a coupling buffer (50 mM Tris-HCl, 5 mM EDTA, TCEP*, pH 8.5) and added to saturation to **UltraLink® iodoclay** resin (**ThermoFisher**, #53155) pre-washed with the coupling buffer.
* pH of TCEP has to be adjusted to 8.0 before.
- Following 1 h incubation at room temperature with mild agitation the resin is settled, washed with coupling buffer and blocked with 50 mM L-cysteine (pH 8.0) for 30 min.
- The AC-coated resin is washed with 1 M NaCl and PBS and stored as 50% (v/v) suspension in PBS containing 2 mM Na₂S₂O₃ at 4°C.

B. Preparation of *E. coli* S30 cell extract (from Schwarz et al.²¹ with minor modifications):

cultivation media TBGG* (1 L)	S30 extract buffers:	S30A* (~2 L)	S30B* (~0.1 L)	S30C* (~8-10 L)
Yeast Extract 24 g	Tris-acetate pH8.2 (mM)	10	10	10
Glycerol 8 ml	Mg(OAc) ₂ (mM)	14	14	14
Glucose 1 g	KCl (mM)	0.6	0.6	-
KH₂PO₄ 2.31 g	KOAc (mM)	-	-	0.6
K₂HPO₄ 12.54 g	2-mercaptoethanol (mM)	6	-	-
	DTT (mM)	-	1	0.5
	BMSF (mM)	-	0.1	-

* prepare x50 stocks for S30AB and S30C (Tris/Mg/K)

- Overnight culture of BL21(DE3) *E. coli* is inoculated into 5 L TBGG media at 1:100 dilution, distributed over 6 conical 5L flasks (~850ml per each) and grown at 37°C with agitation.
- In ~3-4 h, after reaching the log phase, cell culture is pooled and rapidly cooled down with 5x200 ml -80-pre-frozen packs of LB broth to yield 6 L of final culture volume.
- Following complete dissolution of frozen LB packs, cells are pelleted and washed twice with 1L of pre-chilled S30A buffer (every time cells were spun for 15 min at 2.5kg at 4°C).
- The cell pellet after the final wash is resuspended in 2V (vol/vol) of S30B buffer, and cells are disrupted by fluidic continuous flow disruption (Constant Systems CF1) with 20 kpsi at 4°C, followed by two consecutive centrifugation steps of the cell homogenate for 30 min at 30kg at 4°C while collecting top 3/4th of supernatant every time.
- Final supernatant is pooled and adjusted to 0.4 M with 5 M NaCl followed by 45 min incubation at 42° in a water bath.
- Following incubation the cell homogenate is transferred to dialysis tubing with 12-14 kDa cutoff (**Spectrum™** Labs) and dialyzed for 2 h against 4 L of cold S30C buffer at 4°C followed by ON dialysis against the same volume of fresh buffer.
- Following ON dialysis the extract is centrifuged for 30 min at 30kg at 4°C, top 3/4 of supernatant is collected, aliquoted, frozen in LN₂, and stored at -80°.

3. Resin-assisted reaction mixtures are incubated in round-bottom tubes in a thermomixer (Eppendorf) for 4 h at 1050 rpm at 32°C and 28°C for peptide and protein translation, respectively.

4. Following translation reaction, the reaction mixtures are transferred to filter-bottom tubes, flowthroughs are removed and resins are washed by six alternating sessions of either water and WB (refer to the Table) for immobilized peptides or WB and NB for immobilized proteins, followed by three continuous final washes by water and NB, respectively.

D. Downstream refolding:

Common buffer compositions:

label:	buffer name:	buffer composition:
WB	wash buffer (AC-resin)	50 mM TrisHCl, 500 mM NaCl, 0.1% Tween 20, pH 7.5
WB1	wash buffer (ST-resin)	20 mM Tris-HCl, 100 mM NaCl, 0.05% Tween 20, pH 7.5
NB	neutral buffer	20 mM TrisHCl, 20 mM NaCl, pH 7.5

For refolding procedure 40 µl of buffer was used per resin amount corresponding to 10 µl of 50% (v/v) suspension.

Procedure for disulfide-constrained peptides (AC-assisted CFS):

Process:	Conditions:	Time	Agitation (rpm)	T°C
Translation	AC-assisted CFS, standard	4 h	1400	32
Washing x6	WB/NB/WB/NB/WB/NB	na	na	na
Washing x3	AQ	na	na	na
Full reduction	50 mM DTT, 0.1 M NH ₄ HCO ₃ , pH 8.5	30 min	1400	na
Washing x3	0.1 M NH ₄ HCO ₃ , pH 8.5	na	na	na
Oxidative folding	10 mM GSH, 0.1 M NH ₄ HCO ₃ , pH 8.5	12-48 h	1400	RT
Washing x3	0.1 M NH ₄ HCO ₃ , pH 8.5	na	na	na
Washing x3	AQ	na	na	na
Elution x3	0.2% TFA	na	1400	na

Procedure for disulfide-constrained proteins (AC-assisted CFS):

Process:	Conditions:	Time	Agitation (rpm)	T°C
Translation	AC-assisted CFS, (+7.5 mM DTT, S30 extract 1:2)	4	1400	25
Washing x6	WB/NB/WB/NB/WB/NB	na	na	na
Washing x3	NB	na	na	na
Refolding step 1	2 M Gdn, 100 mM DTT, 50 mM TrisHCl, 1M NaCl, pH7.5	1 h	1400	na
Refolding step 2	1 M Gdn, 10 mM GSH, 50 mM TrisHCl, 500 mM NaCl, pH7.5	ON	1400	RT
Dilution 1:2	NB	1 h	1400	na
Dilution 1:2	NB	1 h	1400	na
Washing x3	NB	na	na	na

C. Assembly of AC-resin-assisted *E. coli* CFS reaction

C-1 Preparation of affinity-resin for translation reaction

- The resin amount corresponding to 30-40 µl of 50% (vol/vol) suspension is used for 100 µl of transcription-translation mixture.
- The required volume of 50% resin suspension was transferred into a filter bottom tube, followed with a quick 1kg spin to drain the liquid and with 6 continuous washes with water. Following the final spin at 2.5kg the resin was equilibrated with 1V of the reaction media consisting of 35% (v/v) of extract buffer and 40% (v/v) of x2.5 Feeding solution (refer to the table below) in two consecutive buffer additions: while the first portion is drained by centrifugation at 2.5kg for 1 min, the resin is stored with a second portion of equilibration buffer at RT while transcription-translation mixture is prepared.

C-2 Assembly of transcription-translation reaction

- The reaction constituents are mixed on ice as per the table below: (for translation of disulfide-constrained proteins DTT concentration in translation reaction is adjusted to 10 mM with extra 7.5 mM DTT)

Standard protocol for the assembly of transcription-translation reaction:

Major constituents:	Components:	Concentration:	Reaction assembly:
S30 extract	Tris-acetate pH 8.2	10 mM	35 % (v/v)
	Mg(OAc) ₂	14 mM	
	KOAc	0.6 mM	
	DTT	0.5	
	Hepes-KOH pH7.6	236 mM	
x 2.5 Feeding Solution	Mg(OAc) ₂	12.5 mM	40 % (v/v)
	KOAc	375 mM	
	PEG 8000	5 %	
	Na ₃ N	5 mM	
	Tween 20	0.015%	
	DTT	5 mM	
	cComplete™ EDTA-free, Protease inhibitors (Roche)	x 2.5	
	Ethanol	0.25 mg/ml	
	rNTPs each	2 mM (each)	
	ATP (extra)	1 mM	
	Acetyl phosphate	38 mM	
	Creatine Phosphate	68 mM	
Enzymes	20 amino acids	1.25 mM (each)	0.05 mg/ml
	R.C.W.D.G (extra)	2.5 mM (each)	
Enzymes	T7 polymerase	stock	45 U/ml
	Creatine phosphokinase	stock	0.05 mg/ml
Template	TMV-protease	stock	0.05 mg/ml
	Plasmid DNA	0.2-0.4 µM	20-40 pM

- Following the assembly of reaction mixture the affinity resin is drained off the equilibration buffer for 1 min at 2.5kg and semi-dried resin is directly emptied into the reaction mixture.

Procedure for proteins lacking disulfide bonds (ST-/AC-assisted CFS):

Process:	Conditions:	Incub. time	Agitation (rpm)	T°C
Translation	ST-/AC- assisted CFS, (S30 extract 1:2)	3 h	1050	28
Washing x6	WB1/NB/WB1/NB/WB1/NB	na	na	na
Washing x3	NB	na	na	na
Refolding	1.6 M Gdn, 20 mM TrisHCl, 2.5 mM DTT, 20 mM NaCl, 0.125% Tween 20, pH7.5	2 h	1400	RT
Refolding (AC-resin)	2 M Gdn, 20 mM TrisHCl, 2.5 mM DTT, 1 M NaCl, 0.125% Tween 20, pH7.5	2 h	1400	
Dilution 1:2	NB	1 h	1400	
Dilution 1:2	NB	1 h	1400	
Washing x3	NB	na	na	
Elution x2 (ST-resin)	50 mM biotin, 100 mM Tris-HCl pH 8.5, 150 mM NaCl, 0.25 mM EDTA	5 min	1400	na

Point 3. The results obtained in this work could have been discussed or compared with the current state-of-the-art capabilities of the field. For instance, some companies are selling cell-free systems specifically for producing peptides and proteins with disulfide bonds (e.g., SHuffle kit from the company NEB). Some of the results obtained with a lysate-based E. coli cell-free system are compared to the PURE system which is good. Another area that could have been included in the discussion is the current trend for fast biomanufacturing at the point of care. For example, several groups have shown how to prepare lyophilized cell-free proteins synthesis systems, removing the need for a cold chain and enabling biomanufacturing at the point of care. How does the approach described in this article position itself with respect to that? If the approach is not yet adapted for point-of-care applications, just state it, and explain how it could be adapted for point-of-care applications (if it is possible).

Response to point 3: As for the first point raised by the Reviewer regarding the comparison of our system with the current state of the field, we feel that we previously provided a brief summary in the introduction subchapter in which we discussed the general trend and associated limitations in the field. We also extrapolated to how the currently proposed strategy could address them. We argue that at physiological conditions it is generally hard for translated polypeptides to avoid the kinetics traps resulting in misfolding and aggregation in particular in the cell-free system due to possible loss of concerted chaperon activity. Therefore, majority of studies trying to achieve the productive tradeoff between folding and aggregation through combining the oxidizing environment for the accelerated closure of disulfide bridges with complex chaperon cocktail. The Shuffle strain was developed on purpose to support the disulfide bond closure in bacteria cytoplasm. We indirectly mention the Shuffle T7 strain when discussing the previous attempts to produce Pn3a, the neurotoxin, also belonging to our test set, as we cite the work by Sharma et al. where a panel of strains was screened including the Shuffle T7. Although this strain favorably compared with the rest of expression strains in terms of supporting high yields of soluble product – the peptide of interest was remaining in a largely misfolded state. This is consistent with the fact that despite the notion that disulfide bond closure directs folding by providing the constraints many studies show that accelerated formation of disulfide bonds results in randomly cross-linked intermediates. Therefore, we argue that the success rate of the attempts to achieve this tradeoff would still hinge on individual features of translated sequences while thermodynamically controlled conditions would be universally applicable.

“However, the rest, partially undergoing “collapsed” folding 34, demanded a downstream refolding at optimized redox conditions despite their expression as fusions with MBP in E. coli periplasm or in the engineered E. coli strain promoting oxidative folding^{47, 48}.”

As for the suitability of the proposed strategy for biomanufacturing at the point of care – in our previous response to the Reviewer we acknowledged the importance of this subject as it projected for us to the question of scalability performance of the proposed strategy in general. We concluded that the proposed approach in its current format is limited by the use of the protein-coated affinity matrix which can be recycled up to seven times only with a certain loss of binding capacity following each recycling step. Therefore, we concluded that at its current format the approach would be mostly amenable for prototyping rather than for biomanufacturing and provided the statement in the discussion subchapter:

“Although the proposed strategy is likely to benefit from the integration with homeostatic energy regeneration 6 and the use of engineered strains 8, 36 it is important to note that in its current format it is rather applicable for small-scale prototyping due to limited reuse of the protein-coated resin.”

Since the proposed strategy is not applicable for biomanufacturing in its present format we believe that the same would be true for the point of care application since biomanufacturing scale would be a critical factor. On another hand in the same paragraph, we suggested a

potential route to tailor the approach towards biomanufacturing scale and outlined the possible issues on this way:

“However, it can be potentially tailored to biomanufacturing via utilizing an inexpensive and recyclable metal affinity resin instead. In contrast to protein-coated resin the latter would give an advantage of using elevated denaturant concentrations and prolonged incubation period to ensure complete resolution and recycling of all inter- and intramolecular aggregates. It may come, however, at a cost of compromised product purity^{42, 54} and the need of readjusting the reductant concentration at multiple stages to avoid metal reduction.”

While not being suitable to biomanufacturing the proposed strategy, offering a standardized procedure for prototyping a variety of polypeptides, can be of advantage for the synthetic biology in streamlining the design-build-test cycles, rapid iteration through which is crucial for the synthetic biology progress – we also provided a brief account of this in the discussion chapter:

“Offering a standardized procedure (Supplementary methods) for prototyping of a wide variety of (poly)peptides the proposed strategy can aid in protein/peptide engineering by streamlining design-build-test cycles, a rapid iteration through which remains one of the major bottlenecks of synthetic biology.”

Unfortunately, we cannot further elaborate on the point of biomanufacturing at the point of care as the size of the manuscript is currently hitting its limit and we already had to reduce it by 700 words.

Minor points:

Minor point 1: Using cell-free translation all over the place is confusing, as it appears that the work is cell-free transcription and translation for the most part.

Response: Following the reviewer’s concern raised during the first round of reviewing process we already converted “cell-free translation” to “cell-free transcription-translation” at five instances in the manuscript whenever we refer to the technical side (in Results (2) and Methods (3) sections) while for the introduction and discussion subchapters we reserve a more general term “cell-free translation” (occurs at three instances) as we discuss the concerns linked to translation and we feel that the additional use of “transcription” term would not be practical in this light.

Minor point 2: Methods: should be reviewed and completed. For example, what is the composition of buffer S30B?

Response: We now provided a detailed workflow for the assembly of resin-assisted translation reaction where all buffer compositions are provided in a table format:

B. Preparation of *E. coli* S30 cell extract (from Schwarz et al.⁵¹ with minor modifications):

cultivation media TBGG* (1 L)		S30 extract buffers:			
		S30A*	S30B*	S30C*	
		(~2 L)	(~0.1L)	(~8-10L)	
Trypton	12 g	Components:			
Yeast Extract	24 g	Tris-acetate pH8.2 (mM)	10	10	10
Glycerol	8 ml	Mg(OAc) ₂ (mM)	14	14	14
Glucose	1 g	KCl (mM)	0,6	0,6	-
KH ₂ PO ₄	2,31 g	KOAc (mM)	-	-	0,6
K ₂ HPO ₄	12,54 g	2-mercaptoethanol (mM)	6	-	-
		DTT (mM)	-	1	0,5
		PMSE (mM)	-	0,1	-
* prepare x50 stocks for S30AB and S30C (Tris/Mg/K)